Subject Areas:
geology/engineering geology/civil engineering

Keywords:
tunnel engineering, model test, seepage and weak interlayer, failure modes, failure mechanism

Authors for correspondence:
Jing Hu
e-mail: 347902268@qq.com
Haijia Wen
e-mail: jhw@cqu.edu.cn

# Effects of seepage and weak interlayer on the failure modes of surrounding rock: model tests and numerical analysis

Jing Hu[1,2,3], Haijia Wen[1,2,3], Qilong Xie[2], Binyang Li[2] and Qu Mo[4]

[1]Key Laboratory of New Technology for Construction of Cities in Mountain Area, Ministry of Education, and [2]School of Civil Engineering, Chongqing University, Chongqing 400045, People's Republic of China
[3]National Joint Engineering Research Center for Prevention and Control of Environmental Geological Hazards in the TGR Area, Chongqing University, Chongqing, People's Republic of China
[4]School of Civil Engineering, Chongqing JiaoTong University, Chongqing 400074, People's Republic of China

JH, 0000-0002-2717-7848; HW, 0000-0002-2045-729X

The presence of weak interlayers and groundwater are common adverse geological conditions in tunnels. To investigate the modes of failure of rock masses surrounding tunnels owing to weak interlayers and groundwater, model tests and numerical simulations were conducted in this study based on two cases, and a model that considers only the weak interlayer was conducted for comparison. Based on the tests, differences between two models in terms of rock pressure, displacement, cracks and strain were analysed. The results reveal that the presence of groundwater has a significant effect on the space–time distribution of stress, displacement and cracks in the surrounding rock. Furthermore, based on the numerical model, the seepage field was analysed in terms of pore water pressure, permeability and the seepage process to understand the joint action of groundwater and weak interlayer on the failure mechanism of tunnels. The results show that the groundwater and interlayer complement each other to induce the failure mode of the surrounding rock. The water accelerates slip in the interlayer and the development of cracks. Conversely, low strength, muddy weak interlayers serve as the channels of water flow, resulting in deformations and cracks at different locations and different failure modes.

# 1. Introduction

Mountain tunnels have been constructed at a large scale in China in the last few years. Many challenges, including the presence of groundwater, weak interlayers and other unfavourable geological conditions, are inevitably encountered in this process. Given the geological conditions, this set-up leads to numerous problems and geological hazards in the tunnel that threaten the overall stability of the surrounding rocks during excavation. Therefore, it is of great significance to understand the failure mechanism of tunnels as influenced by the joint action of water and a weak interlayer in theory and in practice. The researches on failure mode of surrounding rock are fruitful [1–7], for the failure modes of that tunnel considers only the weak interlayer or groundwater, as described below.

The weak interlayer, because of its low strength and stiffness, exerts a significant influence on the tunnel's stability. Many failures of underground openings have been reported to be closely related to a weak interlayer nearby [8,9]. Furthermore, zones of the weak interlayers in these cases exhibited a loose structure, weak intergranular bonding and low strength, which led to many stability-related problems, such as the sliding of the body of the Bouzey Dam in France and the failure of the foundation of the Austin Dam in Texas [10]. Based on case studies, Nilsen [11] found that rock fall occurs in areas where a fault zone containing swelling clay intersects with the tunnel. McClure & Horne [12] claimed that adverse geological conditions can increase the induced seismicity hazard. It thus remains challenging to ensure the stability of the tunnel over its planned service life, particularly when it is driven through a weak interlayer [13]. Several researchers have examined ways to prevent instability in tunnels. Wang *et al.* [14] and Yin & Zhou [15] developed a value setting method for shear strength and a formula for the ultimate load by considering the weak interlayer. Wang *et al.* [16] proposed a nonlinear finite-element method to simulate layered rock masses and the weak interlayer. Many scholars [17–26] have also investigated the deformation, fracture process and failure mechanism of rock masses with weak interlayers.

In addition to a weak interlayer, groundwater is an unfavourable geological condition influencing the stability of tunnels. In engineering, tunnels are often constructed in aquifers, such as subsea tunnels, cross-river tunnels and many urban tunnels. Therefore, the destabilizing effect of underground water (seepage forces or excessive pore water pressure) should be considered [27]. Freeze & Cherry [28] revealed that some of the most disastrous experiences in tunnelling have resulted from the interception of large flows of water from highly fractured water-saturated rocks. Groundwater inflows may constitute a hazard as well as an important factor controlling the rate of advancement in driving a tunnel [29]. Thus, the destabilizing seepage force has a severe effect on the stability of the tunnel face, and the effect of seepage on such stability must be considered in engineering design and construction [30]. Many contributions have been made in the literature to this issue. Fahimifar & Zareifard [31] proposed analytical solutions for tunnel analyses below the groundwater level under axisymmetric strain on a plane. The authors in [32–37] investigated the stability of the tunnel face with seepage flow by using an improved method. Dadashi *et al.* [38,39] developed methods to evaluate the permeability of the tunnel and the rate of inflow of groundwater into excavated tunnels.

In conclusion, a lot of researches on failure modes in two undesirable geological conditions have been, respectively, by the predecessors, whereas research on the effect of the coexistence of seepage and a weak interlayer has dealt mainly with slope stability, and scant work has considered this scenario in tunnels. What is more, the coexistence of weak interlayers and groundwater are common in tunnels, and will complicate the geological conditions [40–43], causing the difference of failure modes. Therefore, it is important to understand the mechanical and failure modes of the tunnel under the joint action of water and a weak interlayer to design the requisite support and ensure safe excavation.

In this paper, as a typical mountain tunnel, the Baishiyi tunnel contains abundant underground water crossing through a single, weak interlayer in the Zhongliang Mountains and is used as the object of study in this paper. To investigate the effect of the coexistence of seepage and a weak interlayer on the stability of tunnels, model tests and numerical simulations are conducted. The difference in failure modes of the surrounding rock for two cases are analysed based on results of the model test. Then, the results of experiment and numerical analysis are compared qualitatively and quantitatively to prove the reliability of the numerical simulation method. Subsequently, to investigate the mechanism causing the difference in failure modes, characteristics of the variation in the seepage field are further analysed based on results of the numerical simulation. Finally, the effect of seepage and weak interlayer on failure mechanism was discussed.

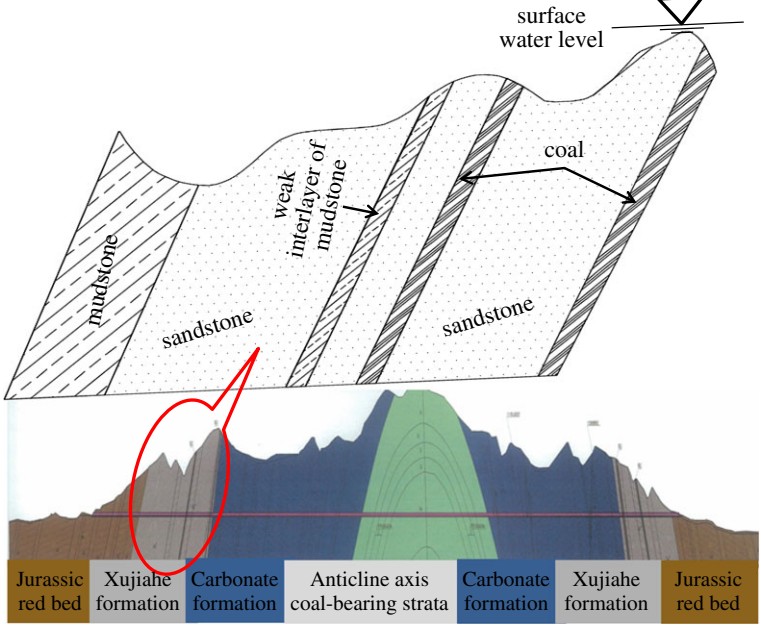

**Figure 1.** Details of the longitudinal fault and research section of the Baishiyi tunnel.

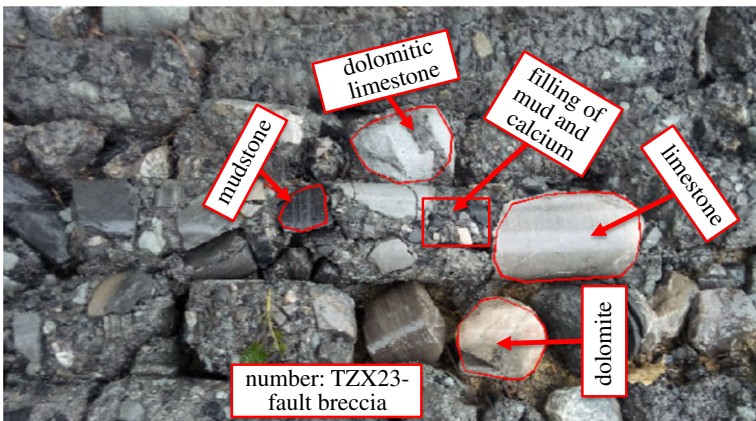

**Figure 2.** Typical photograph of the weak zone.

## 2. Engineering background

As shown in figure 1, the proposed Baishiyi tunnel in Chongqing is a horseshoe-type single-arch double-lane tunnel with a building boundary 13 m wide, a clear height of 9.3 m and total length of 4.6 km. The tunnel crosses the Zhongliangshan anticline, which is a low-mountain landform with tectonic denudation. The terrain is high in the east and low in the west. It is characterized by a terrain slope of 35–50°, ground elevation of 400–550 m and the gully is vertical and horizontal with a cutting depth of 120 m. The surrounding rocks are mainly composed of hard rock, such as sandstone and limestone. While for a weak interlayer, as seen in the typical photograph of weak zone (in figure 2), it is mainly composed of mudstone, mud and calcium, dolomitic limestone and coal series. The rock stratum is developed and interlayer bonding is poor. Groundwater is mainly found in cracks in rocks in the form of limestone karst water. In accordance with the design codes for road tunnels in China, the grade of the surrounding rock and the weak interlayer were IV and V, respectively.

The section chosen to study was the Xujiahe formation in the west wing of the Zhongliangshan anticline through which the tunnel passes. As shown in figure 1, the maximum buried depth of the tunnel was 86.88 m and the surface line was the groundwater level line. As such, the weak interlayer and surrounding rocks were saturated. The physical and mechanical parameters of the surrounding rocks in addition to the weak interlayer were chosen according to the results of a geological survey [44], as shown in table 1.

**Table 1.** Physical parameters of surrounding rocks and weak interlayer.

| original rock material type | density $\gamma$ (kN m$^{-3}$) | elastic modulus $E$ (MPa) | Poisson's ratio $\mu$ | cohesion $c$ (kPa) | internal friction angle $\phi$ (°) | uniaxial compressive strength $R_c$ (MPa) | permeability coefficient $K$ (cm s$^{-1}$) |
|---|---|---|---|---|---|---|---|
| surrounding rock | 24.9 | 9300 | 0.23 | 2160 | 36.5 | 32.3 | $5.1 \times 10^{-6}$ |
| weak interlayer | 22 | 2000 | 0.31 | 120 | 26 | 5 | $4.05 \times 10^{-8}$ |

**Table 2.** General law of similarity of the model text.

| similar constant parameters | relationship | similarity ratio |
|---|---|---|
| geometric similarity constant | $C_l = C_m/C_\rho$ | 45 |
| severe similarity constant | $C_\gamma = C_m/C_\rho$ | 1.16 |
| stress similarity constant | $C_\sigma = C_l C_\gamma$ | 52.2 |
| strain similarity constant | $C_\varepsilon = C_l/C_l$ | 1 |
| internal cohesion similarity constant | $C_c = C_l C_\gamma$ | 52.2 |
| internal friction angle similarity constant | $C_f = C_\mu$ | 1 |
| Poisson's ratio similarity constant | $C_\mu = C_\varepsilon$ | 1 |
| compressive strength similarity constant | $C_{Rc} = C_l C_\gamma$ | 52.2 |
| permeability coefficient similarity constant | $C_k = C_t = \sqrt{C_l}$ | 6.71 |

**Table 3.** Physical parameters of surrounding rocks and weak interlayer in model tests.

| original rock material type | density $\gamma$ (kN m$^{-3}$) | elastic modulus $E$ (MPa) | Poisson's ratio $\mu$ | cohesion $c$ (kPa) | internal friction angle $\phi$ (°) | uniaxial compressive strength $R_c$ (MPa) | permeability coefficient $K$ (cm s$^{-1}$) |
|---|---|---|---|---|---|---|---|
| grade surrounding rock | 21.47 | 178.16 | 0.23 | 41.37 | 36.5 | 0.62 | $3.42 \times 10^{-5}$ |
| weak interlayer | 22 | 38.31 | 0.32 | 2.3 | 26 | 0.096 | $2.66 \times 10^{-7}$ |

# 3. Material and methods

## 3.1. Similarity theory and similar materials

According to the three fundamental laws of the similarity theorem, the ratio of similitude ($C$) can be deduced by means of an analysis of the relevant equation to satisfy the similarity relation, as shown in table 2. The selected geometry and volume–weight similarity ratios were $45:1$ and $1.16:1$, respectively, that is, $C_l = 45$ and $C_\gamma = 1.16$. Consequently, the other similarity parameters were determined as shown in table 2. The similarity material involved two kinds of mixture ratio tests: surrounding rock and weak interlayer. Based on the mechanical parameters (shown in table 1) and the general law of similarity (shown in table 2), the physical parameters of the surrounding rocks and weak interlayer in the model test were deduced, as shown in table 3. The similarity mixture selected for the surrounding rock consisted of sand, white cement, barite powder, talcum powder, silicone oil, petrolatum and gypsum. This has been shown to be appropriate for underground fluid–solid coupling model tests by Li *et al.* [45]. For the weak interlayer, the similarity mixture consisted of clay,

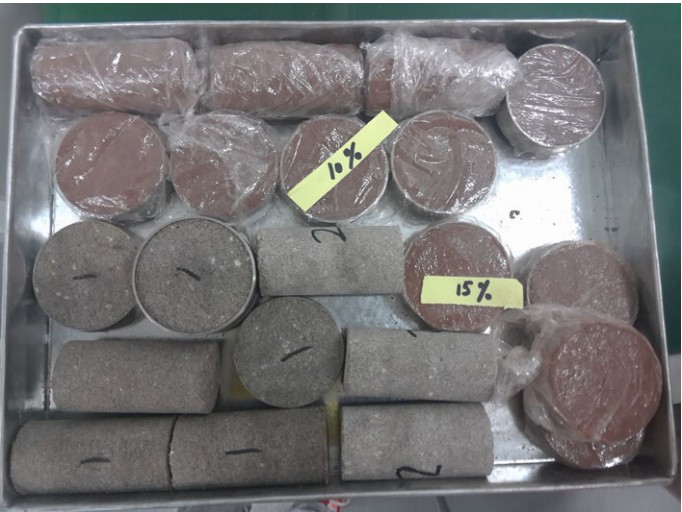

**Figure 3.** Specimens used in material tests.

river sand, talcum powder and silicone oil. It has been shown to be appropriate for the seepage analysis of the weak interlayer [46].

To satisfy the requirements of the model test, various laboratory tests were carried out (as shown in figure 3), including conventional triaxial tests to determine the friction angle and cohesion, dynamic triaxial tests to determine the elastic modulus, unconfined compression tests to measure the uniaxial strength, and permeability tests to achieve permeability coefficient. The ratio closest to the similarity criterion was identified as the mixture ratio. The mixing ratio of the surrounding rocks was as follows: sand : white cement : barite powder : talcum powder : silicone oil : petrolatum : gypsum : water = 48.84 : 19.34 : 5.2 : 7.06 : 1.93 : 1.93 : 2.9 : 12.76. The mixing ratio of the weak interlayer was as follows: clay : river sand : talcum powder : silicone oil : water = 10.6 : 25.5 : 46.3 : 15.6 : 1.9.

## 3.2. Apparatus and measurement of model test

According to the similarity constant $C_l$ and the size of the prototype tunnel, the model tests were conducted in a model box with dimensions $1.6 \times 1.3$ m in a plane, and was 0.4 m thick. As shown in figure 4, the front and back sidewalls of the tank were constructed using transparent Plexiglas plates for observing strain around the tunnel during testing. The transparent Plexiglas plate was 20 mm thick, stiff enough to resist the deformation caused by excavation. To simulate the excavation, holes in the shape of the cross-section of a tunnel were cut on both transparent Plexiglas plates. Steel plates with a thickness of 10 mm were welded to the left and right sides in addition to a third steel plate 20 mm thick at the bottom. Finally, a steel roof plate 15 mm thick was used to convert the concentrated load transmitted by a hydraulic jack into uniform load.

A device was specially designed to simulate tunnel excavation, and its dimensions are shown in figure 5. Some wood bars of small diameter were used to assemble into the tunnel's cross-sectional shape, which was then wrapped in a thin sheet of iron. During tunnelling, the device was pulled out in eight steps.

The configuration of the measuring instruments is shown in figure 6a,b. Pressure from the surrounding rock and pore water pressure were measured. For two cases, a model XY-TY10H earth pressure cell and model XY-SYJ02C osmometers with the same measurement limit of 500 kPa were used to monitor the surrounding rock pressure and pore water pressure, respectively. A DH3816 data collection box was used to collect data monitored during excavation. Note that only an increment in the surrounding rock pressure and pore water pressure was recorded, where a positive value implied an increase and a negative one a decrease. Figure 6c,d shows the distribution of the earth pressure cell and osmometers in the surrounding rocks. To reduce disturbance due to excavation, the buried depth of the earth pressure cell was 10 cm in the radial direction and 15 cm in the longitudinal direction from the cavern. The position of the buried seepage pressure meters was 10 cm in the radial direction and 25 cm in the longitudinal direction from the cavern. Points D1–D8 in all figures are monitoring points located at the vault, left spandrel, left haunch, left arch springing, arch bottom, right spandrel, right haunch and right arch springing, respectively. Similarly, points J1–J7 are monitoring points located at the weak interlayers.

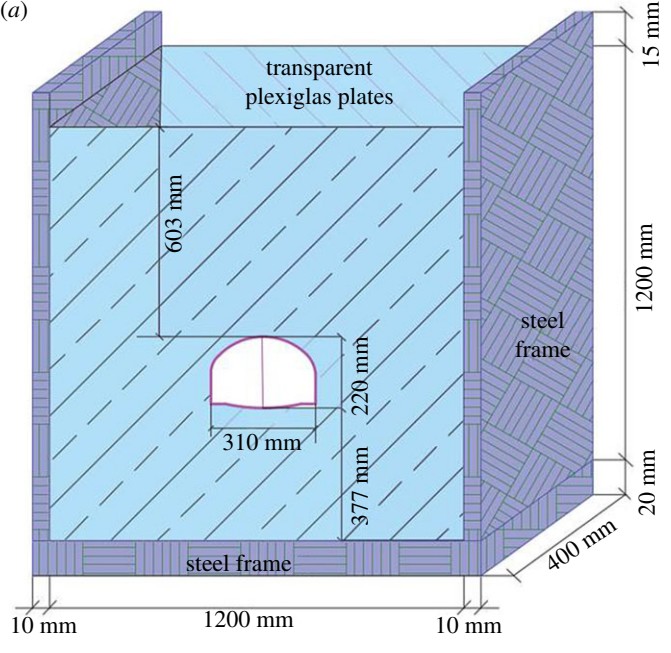

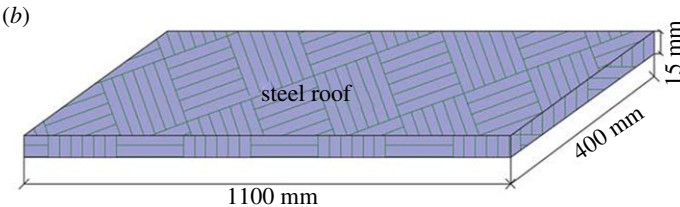

**Figure 4.** Diagram of the model box (unit: mm). (*a*) Model box and (*b*) cover plate of the model box.

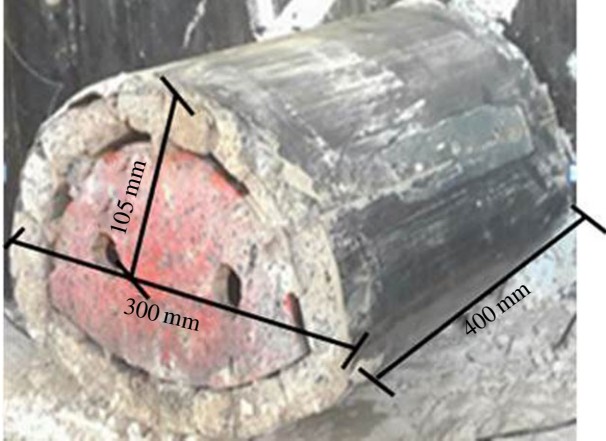

**Figure 5.** Tunnel excavation device (unit: mm).

In addition, the VIC-3D7 [47] speckle technique was used to monitor the vertical, horizontal and shear strain as well as the failure mode of the surrounding rocks during excavation. The VIC-3D7 [47] speckle technique and loading device are shown in figure 7.

## 3.3. Procedure of model test

Two types of tunnel models, a tunnel containing a weak interlayer at an inclination angle of 40° and another containing a weak interlayer at an inclination angle of 40° and abundant groundwater, were considered as cases 1 and 2, respectively, and are shown in table 4. Because the only difference

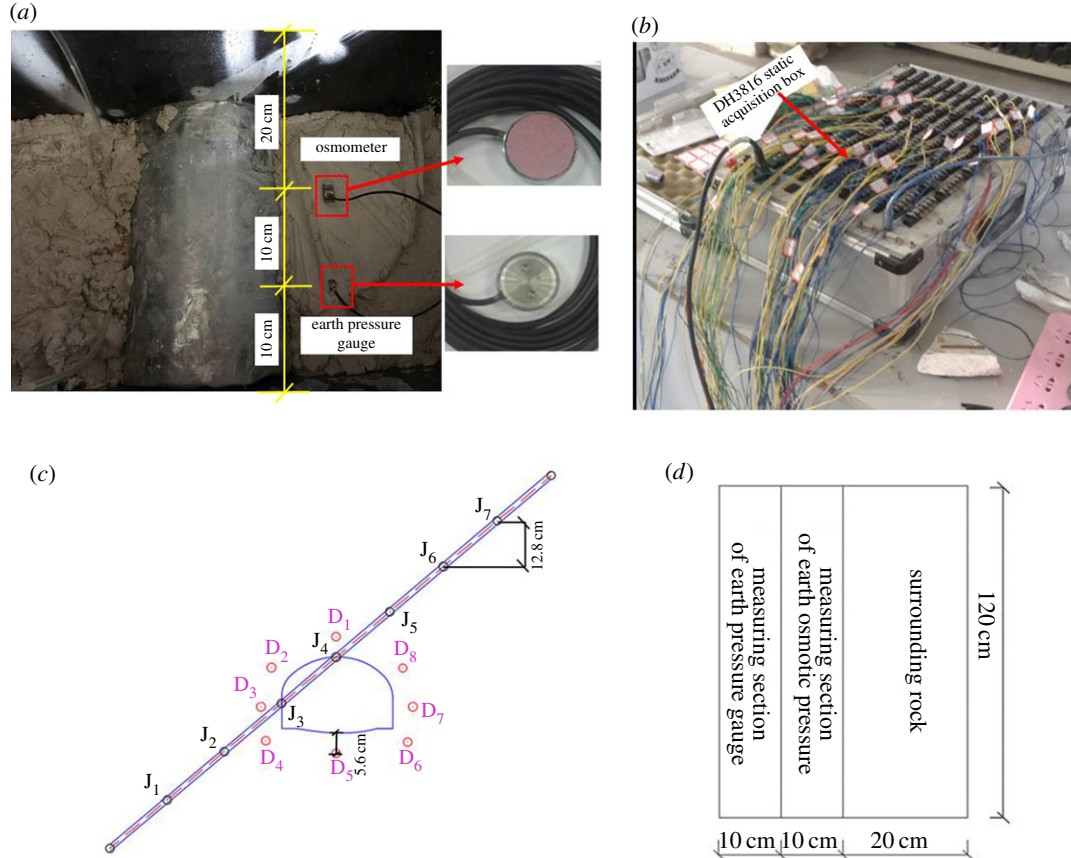

**Figure 6.** Monitoring elements and their distribution. (a) Embedment of element, (b) DH3816 static acquisition box, (c) transversal distribution of monitoring points and (d) longitudinal distribution of monitoring points.

between the models was the presence of water, so their spatial distributions were the same, as shown in figure 8, and the tunnel model was horseshoe shaped with a span of 0.3 m and height of 0.21 m. All the boundary conditions of model tests were normal constrained, the surface of the model was set as a free surface and this was the same for water.

To ensure that results of the model tests were consistent with the empirical results, the full-section excavation method was adopted to simulate the excavation. To simplify the test, the excavation process was divided into eight steps. For each step, the mould was correspondingly pulled out by 5 cm.

As shown in figure 7, the loading system at the top was used to reproduce an earth overburden load according to the unit weight of overburden at a certain height. The horizontal stress was reproduced through the counteractive force on the steel frame of the right and left sidewalls. Given that the depth of the tunnel was 86.88 m (as shown in figure 8), a variety of overburden depths were obtained by loading on the top of the model, and this can be calculated using the following equation:

$$H = \left(\frac{P}{\gamma} + 0.608\right)C_l \approx 1.837\mathrm{e} - 3P + 27.36. \tag{3.1}$$

In the above, with respect to the prototype values, $H$ is the overburden depth (m), $\gamma$ is the unit weight of the earth's material (kN m$^{-3}$), 27.36 m corresponds to the given overburden depth in the model and $P$ is the additional load required (kPa). Note that all values in the model tests, including displacement and stress, were converted into prototype values according to a reduced scale (table 2) for comparison with the results of the numerical simulation. The test process was as follows:

Step 1. First, the transparent Plexiglas plates were replaced with steel plates, and the steel plate in the front of the model box was divided into three sections for filling. Second, the steel plates were coated with a layer of oil to reduce friction between the model and the model box, and a layer of gravel was laid on the bottom of the model box to help the maintenance of model. Finally, the positions of the buried monitoring elements were demarcated on the steel plate at the rear, as shown in figure 9a.

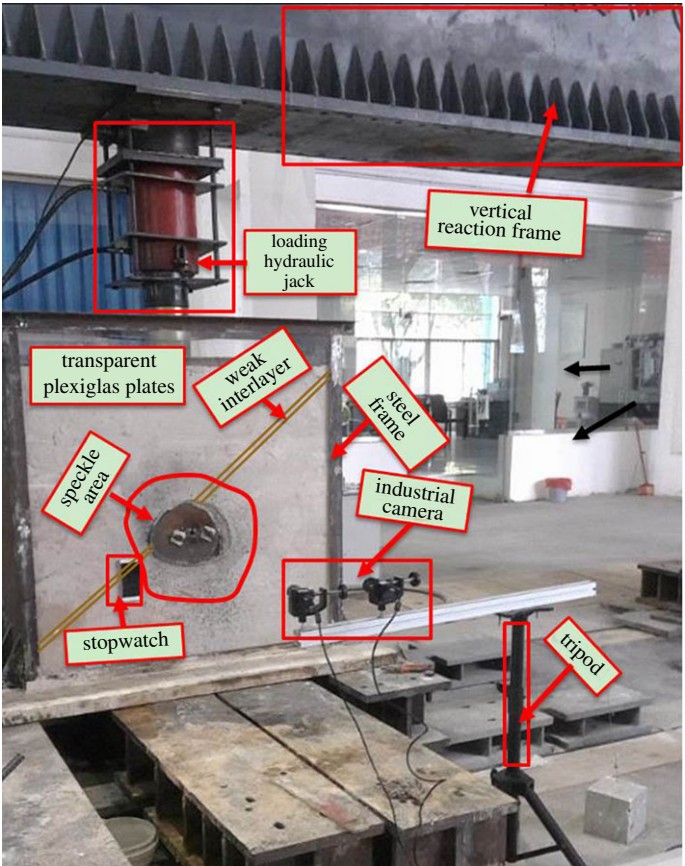

**Figure 7.** Loading device.

**Table 4.** Details of two conditions.

| case 1 | case 2 |
|---|---|
| containing single weak interlayer with a dip angle of 40° and a thickness of 0.015 m | containing single weak interlayer with a dip angle of 40° and a thickness of 0.015 m |
| no groundwater | abundant groundwater |

Step 2. First, the materials were weighed according to a similar mix ratio and evenly mixed in a blender, as shown in figure 9b. Second, the height of paving of the model's ground was 10 cm, and the surrounding rock and weak interlayer were paved layer by layer as shown in figure 9c. Considering the importance of interface, a mixture of sand and clay was used as spacer between the weak interlayer and the surrounding rock. The compacted model ground was sampled to test for suitability. In addition, the earth pressure cell, tunnel excavation device and injection pipe were buried in the corresponding position as shown in figure 6a.

Step 3. First, after maintenance for 36 h to ensure that the model was no longer deformed, the front and back templates were removed. Second, the model was covered with a thin film and maintained for 12 h until the required strength was attained.

Step 4. In the region 15 cm around the cavern, a white primer was sprinkled. After an hour, spots of black paint were evenly sprinkled on it to serve as speckles to acquire camera images. After maintaining this for an hour, the transparent Plexiglas plates on the front and back sidewalls were inserted into the channel steel of the model box.

Step 5. This step was designed for case 2. To saturate the model with water, glass glue was used to seal the gap in the Plexiglas between the surrounding rock and the entrance to the tunnel, as shown in figure 9d. Then, water was injected through the injection pipe as shown in figure 9c. This was

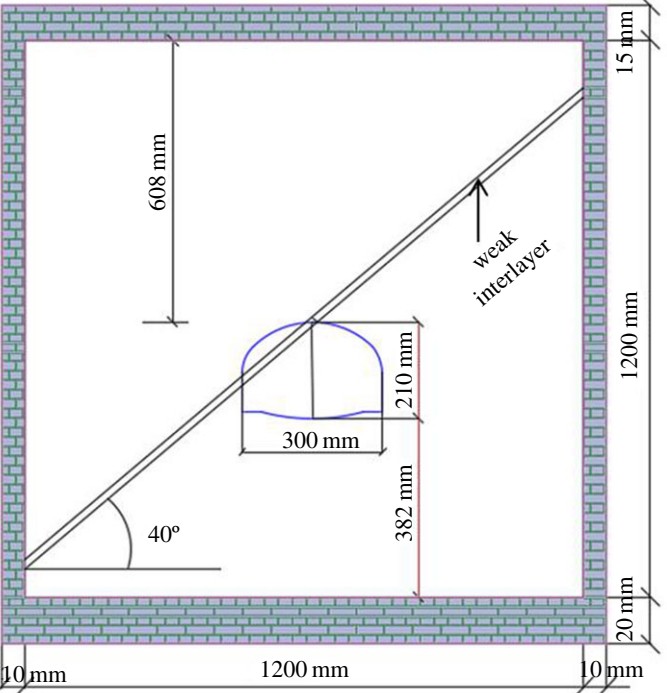

**Figure 8.** Distribution of weak interlayer (unit: mm).

repeated until water seeped out from all sides of the model. The apparatus was maintained for 48 h to attain saturation [19,48,49].

Step 6. Two industrial cameras were placed at the optimal focal length and the speckle was calibrated, as shown in figure 9e. The loading device and information acquisition box were in place. Then, an equivalent overburdened load of 32.4 kN calculated by equation (3.1) was gradually added. Subsequently, once the system was in a stable state, the tunnel excavation device was pulled from the tunnel in eight steps to simulate its full-section excavation [50], where the length of extraction was 5 cm, as shown in figure 9f. After each step of the excavation, the entire system was in a stable state, then data acquisition was carried out.

Step 7. When the excavation was complete, an external uniform load was gradually applied to the ground surface of the model until its structure had been damaged. During this process, the failure mode of rock surrounding the tunnel was observed and recorded.

## 3.4. Numerical analysis

In this study, FLAC3D was used to simulate the excavation process for the aforementioned conditions. The rock mass and the weak interlayer were all Mohr–Coulomb constitutive. Given that the weak interlayer passed through the vault, contact between the upper and lower surfaces, and the surrounding rocks was simulated by the interface unit of FLAC3D. The thickness of the interlayer was 0.675 m, and ran through the entire model. The inclination angle was 40°. In the numerical simulations, the longitudinal length of the tunnel was assumed to be 18 m, and 27 m, along the centreline of the tunnel to the two sides, was used as the region of calculation. Hexahedral, wedge-shaped and columnar grids were used to create the basic grid unit of the model. The total number of units was 21 231 and the total number of nodes was 24 280. A diagram of the numerical model grid is shown in figure 10. All boundary conditions of the numerical model were identical to those of the physical model tests. The constraint along the X-direction was applied to both the left and right boundaries, and that along the Y-direction was applied to the front and rear boundaries. Vertical constraints along the Z-direction were applied to the bottom of the model, the surface of the model was set as a free surface and this was the same for water.

The physical and mechanical parameters were the original rock parameters shown in table 1. The excavation method and process were consistent with the test. For comparative analysis of the results

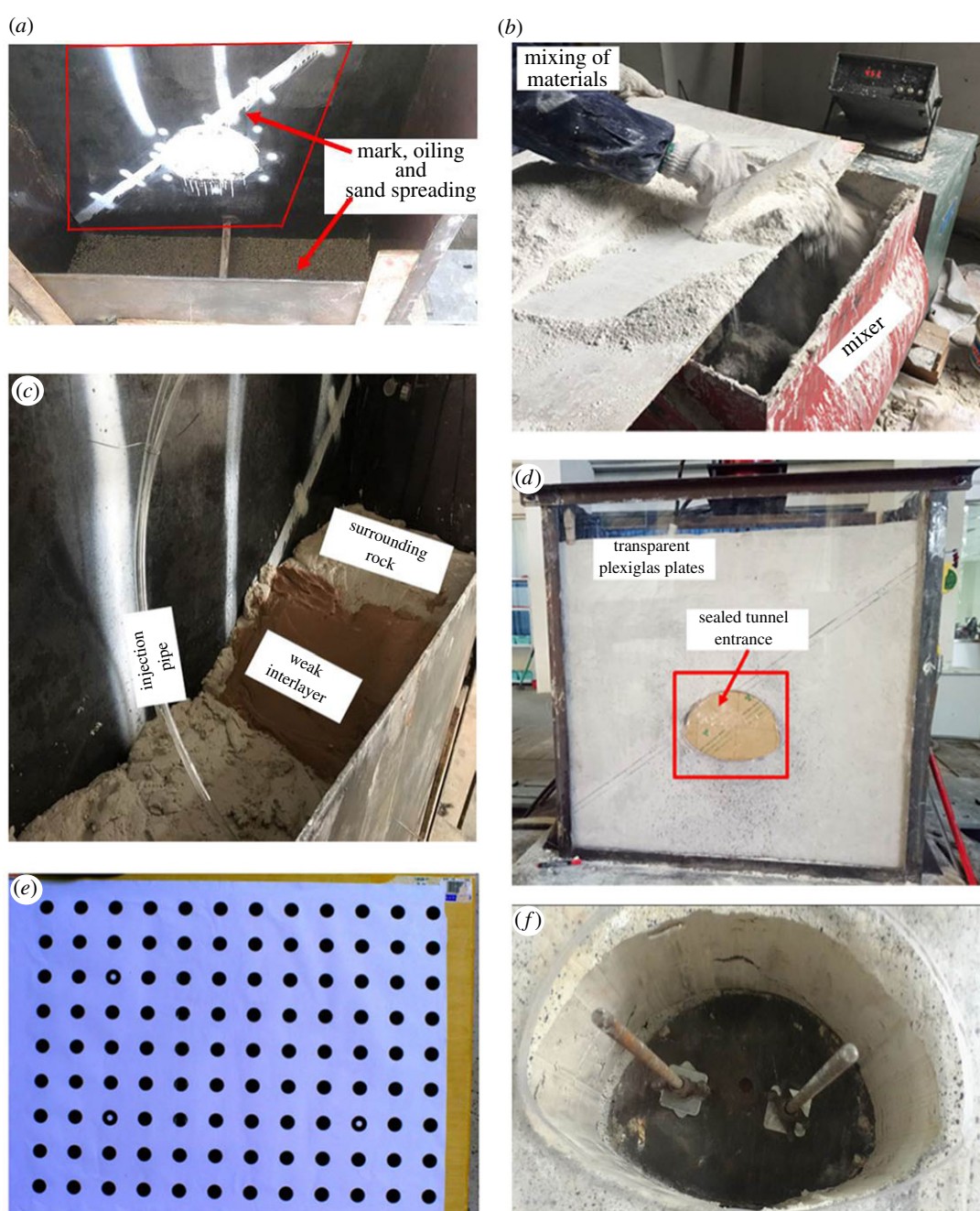

**Figure 9.** Diagram of test process. (*a*) Mark, oiling and sand spreading, (*b*) mixing materials, (*c*) layered paving of materials, (*d*) sealing of the model box, (*e*) calibration and (*f*) tunnelling.

in model tests and numerical simulation, the former should be multiplied by similarity constants according to similarity theory.

In FLAC3D calculations, the coefficient of permeability of the original rock needed to be multiplied by $1.02 \times 10^{-6}$. To monitor dynamic changes in permeability during excavation, the relationship shown in equation (3.2) in FISH language was incorporated into the software so that dynamic changes to the permeability of the medium [38,51] could be monitored in the excavation process. $k_0$ is the original permeability, $\phi_0$ is the initial porosity, %, and $\varepsilon_v$ is volumetric strain

$$k = k_0 \left[ \left( \frac{1}{\phi_0} \right) (1 + \varepsilon_v)^3 - \left( \frac{1 - \phi_0}{\phi_0} \right) (1 + \varepsilon_v)^{-(1/3)} \right]^3. \tag{3.2}$$

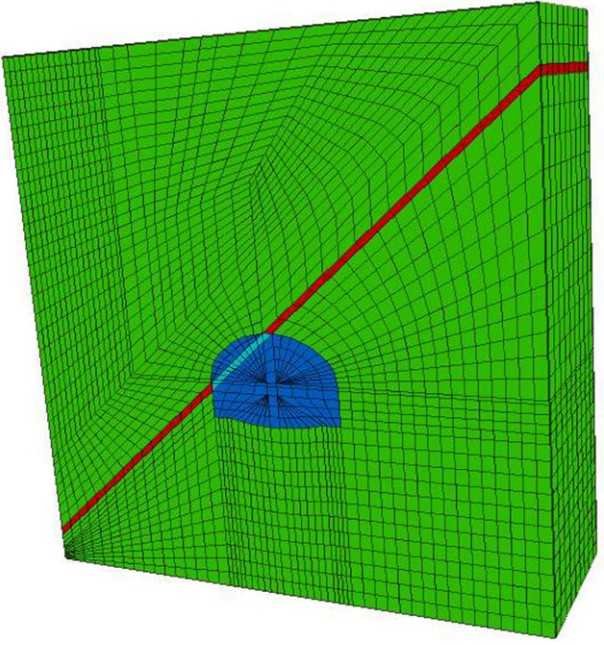

**Figure 10.** Model grid.

# 4. Results of failure modes

In this section, the difference in failure modes of the surrounding rock for cases 1 and 2 are analysed based on results of the model test. Characteristics of the surrounding rocks pressure, displacement field and progressive failure model for two cases are compared and analysed during excavation. Then, the results of experiment and numerical analysis are compared qualitatively and quantitatively to prove the reliability of the numerical simulation method. Finally, to investigate the mechanism causing the difference in failure modes, characteristics of the variation in the seepage field are analysed in §5.

## 4.1. Differences in surrounding rocks pressure

The values of increment in the radial surrounding rock pressure were obtained by monitoring points D1–D8 in the two model tests. Note that in figure 11, subscripts '1' and '2' represent cases 1 and 2, respectively, and the results are described below.

The surrounding rock was divided into a hanging wall and a footwall by the weak interlayer, which resulted in uneven deformation and asymmetric pressure distribution of the surrounding rock in the two cases. Although the results were consistent with past research on the stability of tunnels affected by weak interlayers [19], the coexistence of water and the weak interlayer aggravated the asymmetric distribution of pressure, which endangered the stability of the tunnel.

In case 2, by contrast, the positions vulnerable to failure were mainly distributed around the vault, both kinds of arch springing and spandrel, whereas for case 1, they were mainly distributed around the vault and both haunches, consistently with past research [20]. With the progression of excavation, in case 2, the release in pressure at points D2 (left spandrel) and D4 (left arch springing) changed abruptly. When the excavation had been completed, the value of pressure release at points D2 and D4 accounted for 34.1% and 51.7% of the total release value, respectively, indicating that the left arch springing gradually transitioned into the loose zone [20], and the left invert arch began to work [50]. This is different from case 1, where the release of pressure mainly occurred at left haunch. By comparison, pressure concentration mainly occurred at points D6 (right arch springing) and D7 (right haunch) in case 1, whereas in case 2, under the joint action of water and the weak interlayer, it mainly occurred at points D6 (right arch springing) and D8 (right spandrel). The variation in D8 accounted for 59% of the total pressure value.

Therefore, for case 2, the position of D2 and D4 might have entered the loosening zone, whereas D6 and D8 might have been damaged due to stress concentration. Thus, in terms of practical design and

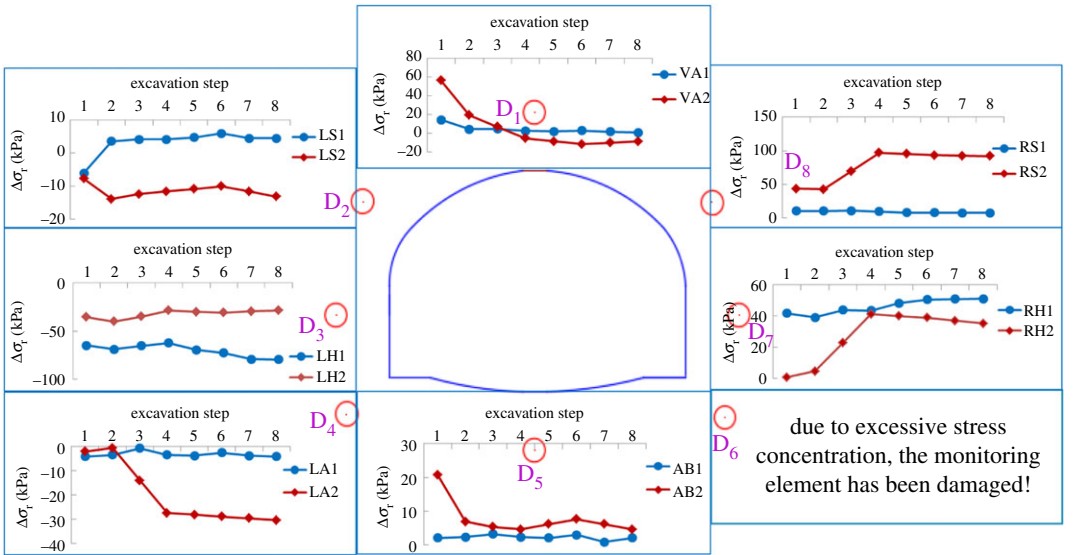

**Figure 11.** Variations in increment in radial pressure in the excavation step in the two cases ($\Delta\sigma_r$, radial increment in surrounding rock pressure; VA, vault; LS, left spandrel; LH, left haunch; LA, left arch springing; AB, arch bottom; RS, right spandrel; RH, right haunch; RA, right arch springing).

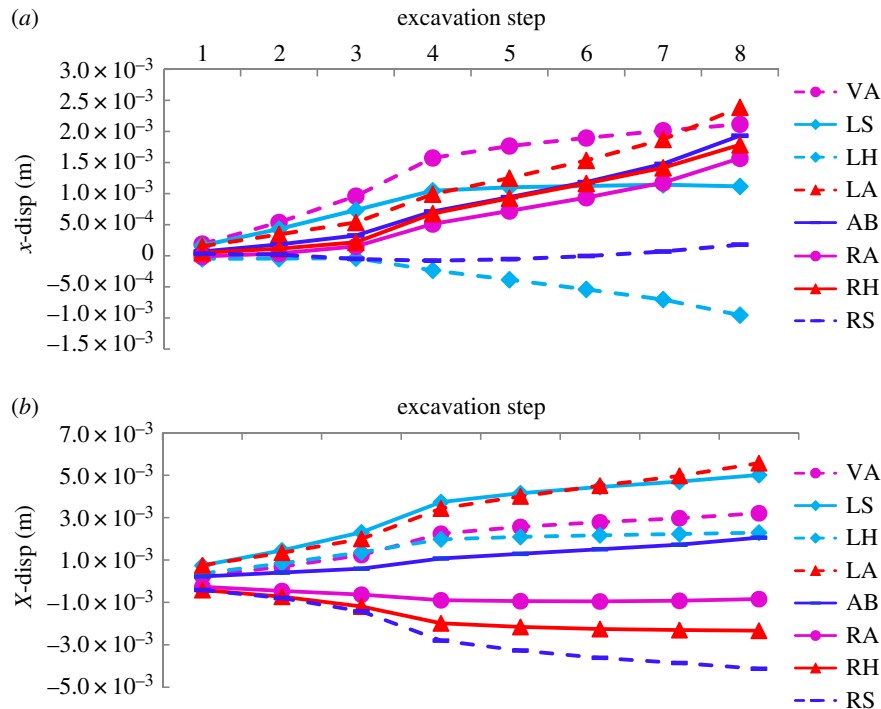

**Figure 12.** Horizontal displacement curves. (*a*) Horizontal displacement curve for case 1 and (*b*) case 2 (VA, vault; LS, left spandrel; LH, left haunch; LA, left arch springing; AB, arch bottom; RS, right spandrel; RH, right haunch; RA, right arch springing).

construction, deformation at the vault, the arch springings and spandrels should be monitored to allow for timely treatment.

## 4.2. Differences in displacement field

Changes in the horizontal and vertical displacement fields of the tunnel in the two cases were simulated and analysed. Their relationships with the excavation step are presented in figures 12*a,b* and 13*a,b*, respectively.

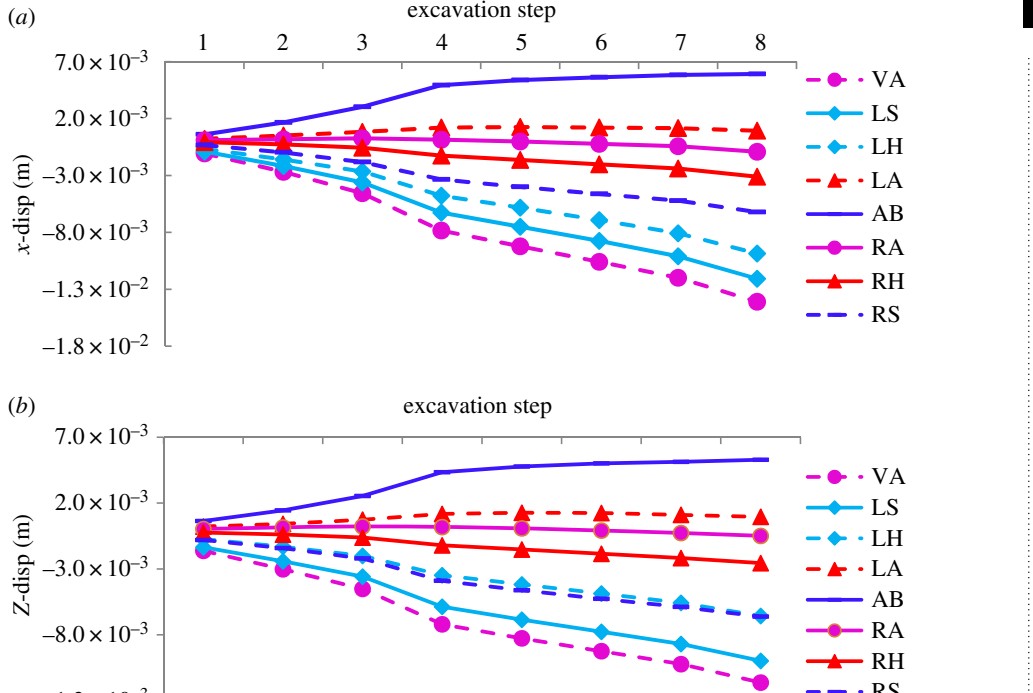

**Figure 13.** Vertical displacement curves. (*a*) Vertical displacement curve for case 1 and (*b*) case 2 (VA, vault; LS, left spandrel; LH, left haunch; LA, left arch springing; AB, arch bottom; RS, right spandrel; RH, right haunch; RA, right arch springing).

**Table 5.** Maximum values of horizontal displacement for two cases (unit: mm).

| case | vault | left spandrel | left haunch | left arch springing | arch bottom | right arch springing | right haunch | right spandrel |
|------|-------|---------------|-------------|---------------------|-------------|----------------------|--------------|----------------|
| 1 | 2.12 | 1.14 | −0.95 | 2.39 | 1.93 | 1.57 | 1.78 | 0.179 |
| 2 | 3.12 | 5.02 | 2.3 | 5.58 | 2.06 | −0.95 | −2.33 | −4.13 |

For variations in horizontal displacement, the maximum values in the two cases extracted by the numerical simulation are shown in table 5. By contrast, differences in horizontal displacement in the two cases were significant. Specifically, in case 2, the maximum values of horizontal displacement at the left spandrel and left arch springing were larger than those at other points, 4.4 and 2.3 times those in case 1, respectively. The maximum values at the right arch springing, haunch and spandrel were rightwards in case 1, whereas in case 2, they turned leftwards. Further, the leftwards horizontal displacement of 4.13 mm at the right spandrel was the largest, which means that both sides of the spandrel and the left arch springing should be considered and provided timely support during excavation.

For variation in vertical displacement, the maximum values were extracted and are shown in table 6. The table and figure 13*a*,*b* show that the negative values represent downwards vertical displacement and positive values represent upwards vertical displacement. The maximum vertical displacement around the cavern in case 2 was slightly smaller than that in case 1. Points with larger settlement, −11.6, −9.99 and −6.62 mm, were at the vault and both spandrels, respectively, and need to be attended to, as in case 1.

The horizontal displacement field in the model test and the numerical simulation for case 1 were compared quantitatively. As shown in figure 14, the law of change in the displacement of the numerical simulation agreed well with those of the model test. Because of friction on the sidewall of the model box, the overlying load could not be transmitted smoothly, causing the peak values of curves in the model test to be smaller than those in the numerical simulation.

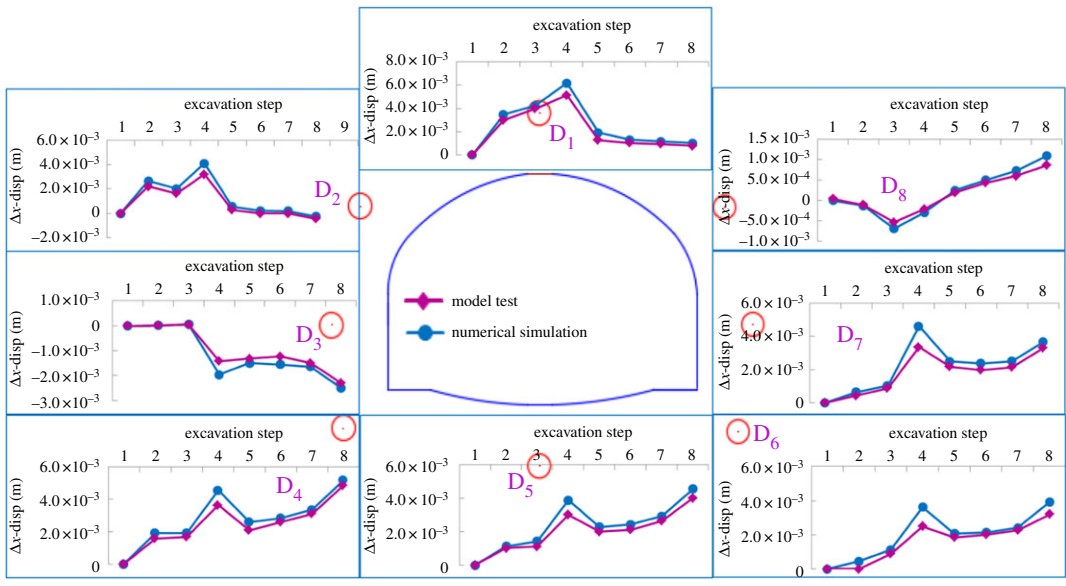

**Figure 14.** Results of horizontal displacement in increment of numerical simulation and model test.

**Table 6.** Maximum values of vertical displacement for two cases (unit: mm).

| case | vault | left spandrel | left haunch | left arch springing | arch bottom | right arch springing | right haunch | right spandrel |
|---|---|---|---|---|---|---|---|---|
| 1 | −14.1 | −12.1 | −9.85 | 1.27 | 5.92 | −0.915 | −3.11 | −6.21 |
| 2 | −11.6 | −9.99 | −6.56 | 1.27 | 5.29 | −0.482 | −2.56 | −6.62 |

## 4.3. Differences in progressive failure model

The progressive process during excavation for cases 1 and 2 can be divided into four stages, as presented in figures 15a–d and 16a–d. The results are given below.

In the initial excavation stage, for case 1, as shown in figure 15a, due to the slippage of the interlayer, the footwall lost support and moved downwards. Meanwhile, shear tension strain occurred at the left spandrel, and horizontally leftwards strain occurred at the left haunch, causing the leftward movement of the hanging wall that formed cracks ① and ② at the vault and the left haunch, at the intersection of the interlayer and the tunnel. Then, horizontal rightwards strain occurred at the right arch springing with the downward movement of the footwall under the action of tension to form crack ③.

In case 2, as shown in figure 16a, the important characteristic was that the temporal and spatial development of crack ② and the deformation mainly occurred in the footwall. This is different from that in case 1. First, the formation of crack ① at the vault was the same as that in case 1. Then, with continuing movement downwards of the footwall, under the action of shear compress, upward vertical strain occurred at the right arch springing near the inverted arch, forming two cracks ②. One of them developed along the right inverted arch, owing to the rebound of the arch bottom, and the other developed along the radial direction owing to the action of the shear compress concentration. According to the pressure distribution shown in figure 11, it is evident that the right arch springing failed to work and the inverted arch began to work.

In the middle and later periods of the excavation stage, as shown in figure 15b, for case 1, vertical strain at the vault decreased gradually with further slippage of the weak interlayer, and turned downwards, indicating that the hanging wall began to move downwards. Then, shear tension strain at the left spandrel turned into shear compress strain, and the rightward horizontal strain zone at this position increased. Then, under the action of shear compress at the left inverted arch, the rightward horizontal strain zone at the bottom of the arch increased, accompanied by a small amount of upward vertical displacement to form crack ③ at the bottom of the arch. Secondary micro-cracks appeared at the right arch springing and the inverted arch.

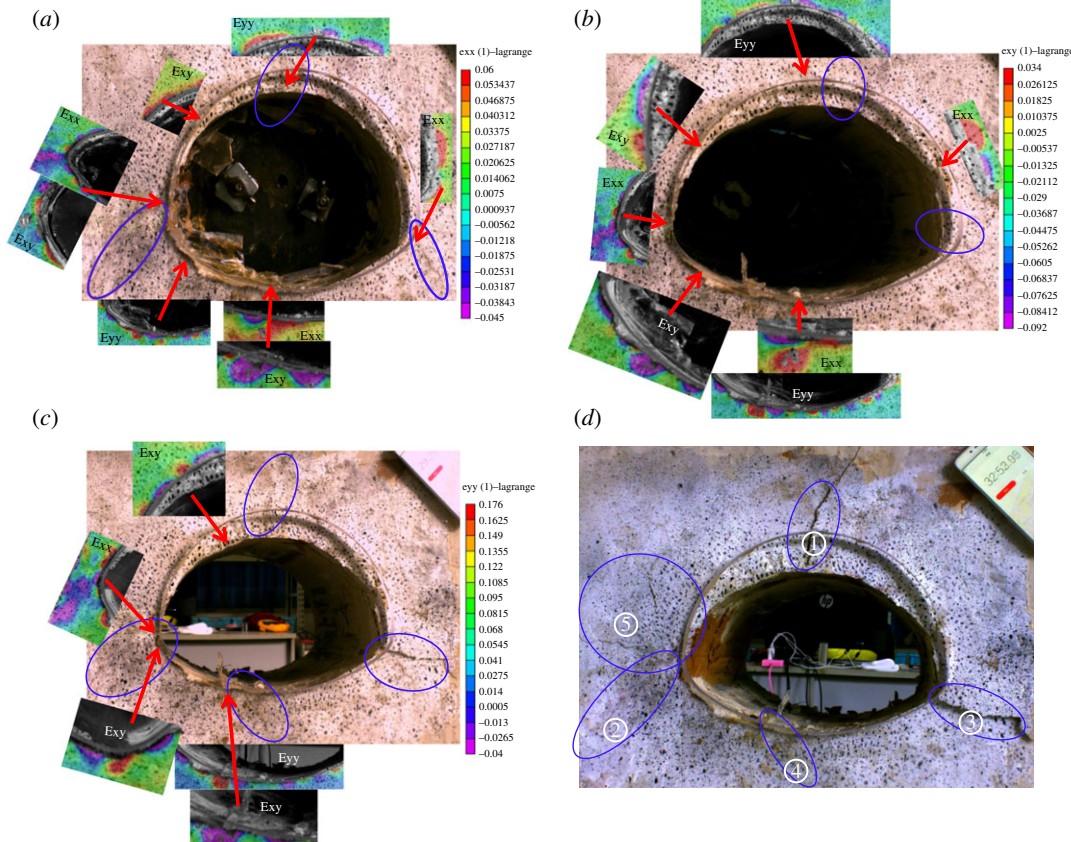

**Figure 15.** Failure process of tunnel only with weak interlayer. (*a*) The crack and strain in the initial excavation stage, (*b*) middle and late excavation stages, (*c*) completed excavation stage and (*d*) step loading stage.

In case 2, in the middle and late periods of the excavation stage, the law of development of cracks and strain is as presented in figure 16*b*. Compared with case 1, the major difference was that the deformation of the left arch springing occurred mainly owing to the combined action of the downwards movement of the hanging wall and the action of shear tension at the right inverted arch, which resulted in a downwards vertical strain at the left arch springing so that crack ③ occurred at this position and developed along the left inverted arch. Subsequently, with the rightwards horizontal strain at the left inverted arch, micro-crack ④ appeared at the bottom of the arch.

When the excavation was completed, for case 1, as shown in figure 15*c*, the zone of shear tension strain at the left spandrel decreased, as did the upwards vertical strain at the arch bottom, and was replaced by the partial downwards vertical strain under the action of shear tension to form shear compress crack ④. Subsequently, owing to the leftwards horizontal stain, the left arch springing was embedded into the rock mass, and secondary micro-cracks ⑤ were formed at the left haunch near the main crack. Finally, micro-cracks on the left haunch and right arch springing were connected with their main cracks. The deformation was then stable due to the re-encounter between the hanging wall and the footwall to form a new stable arch.

In case 2, as shown in figure 16*c*, when the excavation had been completed, upwards vertical and leftwards horizontal strain occurred at the vault near the hanging wall, and a small amount of shear tension and rightwards horizontal strain occurred at the right spandrel to form micro-crack ⑤. In contrast with case 1, this indicates that the interlayer near the vault was still sliding until the excavation of the tunnel had been completed. Finally, a new equilibrium state was reached after the collapse of the vault.

After step loading, as shown in figures 15*d* and 16*d*, the cracks continued to develop, and the main cracks ② and ③ in case 1 were distributed in the left haunch and right arch spring, and presented in a 'reversed V' shape. In case 2, however, they distributed along the inverted arch of both sides of the tunnel in a 'V' shape.

The plastic state of the tunnel in case 2 obtained by numerical simulation was divided into four stages according to the law of development during excavation, to compare with results of the model test as shown in figure 17.

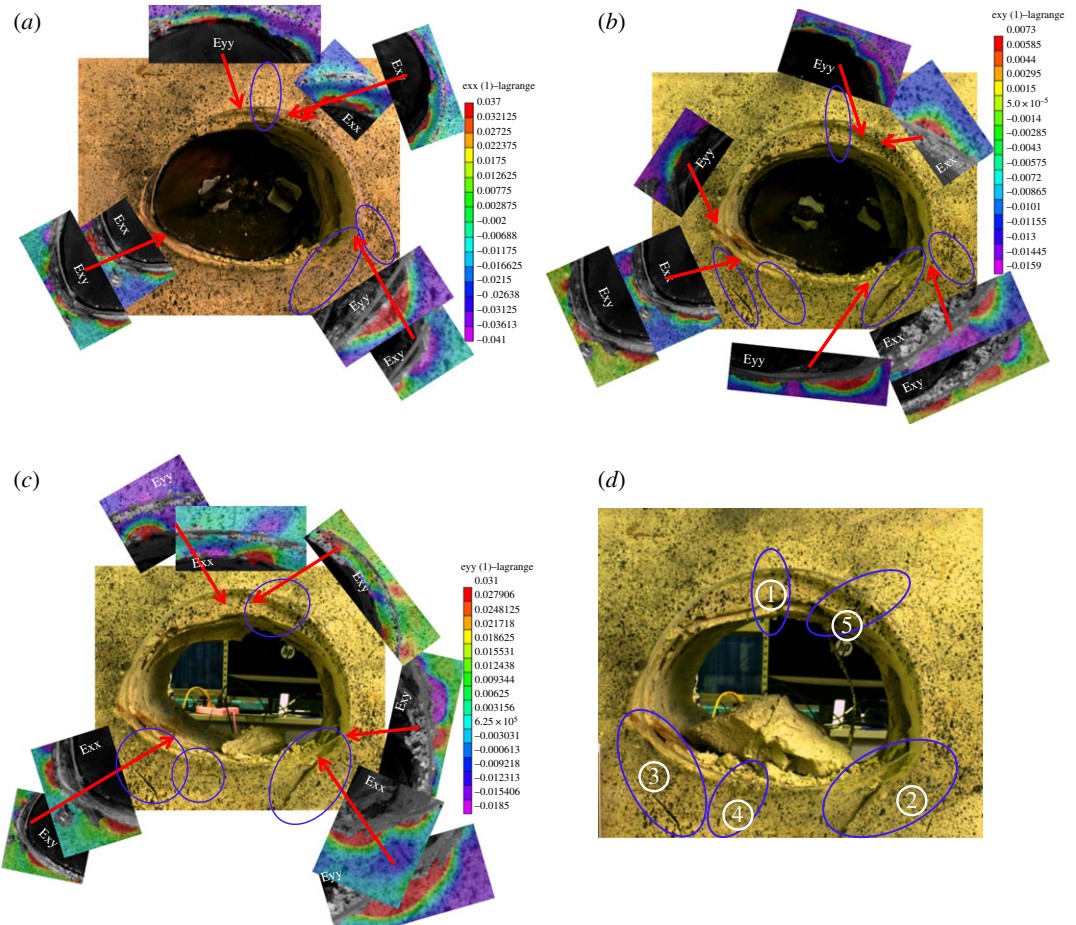

**Figure 16.** Failure process of tunnel with both groundwater and weak interlayer. (*a*) The crack and strain in the initial excavation stage, (*b*) middle and late excavation stages, (*c*) completed excavation stage and (*d*) step loading stage.

In stage 1, when excavation in the third step had been completed, the weak interlayer near the vault failed under the action of shear stress. The zone of shear tension plastic yield in the weak interlayer near the left haunch disappeared, which indicates that only the weak interlayer above the vault slid in this stage, forming shear crack ①, which is consistent with the results of mode tests on case 2 shown in figure 16*a*.

In stage 2, when excavation in the fourth step had been completed, the failure of the interlayer was mainly caused by the action of the tension shear. Because of the substantial pressure release, the left arch springing transitioned into a tension-shear plastic yield state. The zones of the tension plastic yield in footwall also continued to develop and connect, forming cracks ② and ③ in figure 16*b,c*.

In stage 3, that the failure of the interlayer was caused by the action of the shear indicates that the former was still sliding when the excavation of the seventh step was completed. Then, the zone of yield of the tension at the bottom of the arch continued to develop, and plastic zones around the tunnel maintained stable growth. Finally, crack ④, consistently with the result in figure 16*c*, appeared at the bottom of the arch.

In stage 4, when the excavation of the tunnel was complete, the interlayer near the vault turned into a plastic yield state, whereas that near the left haunch was still in the shear plastic yield state, which indicates that slippage continued in this position. The zone of the right spandrel to the right arch springing turned into a tension-shear yielding state to form crack ⑤, which is consistent with the results in figure 16*d*. It is clear that the results obtained by numerical simulations were in good agreement with the laws of change of cracks observed in the model tests.

# 5. Analysis of seepage field

According to the above results, differences in the failure modes of the surrounding rocks, and displacements and developments in the strains and cracks were analysed. The existence of water

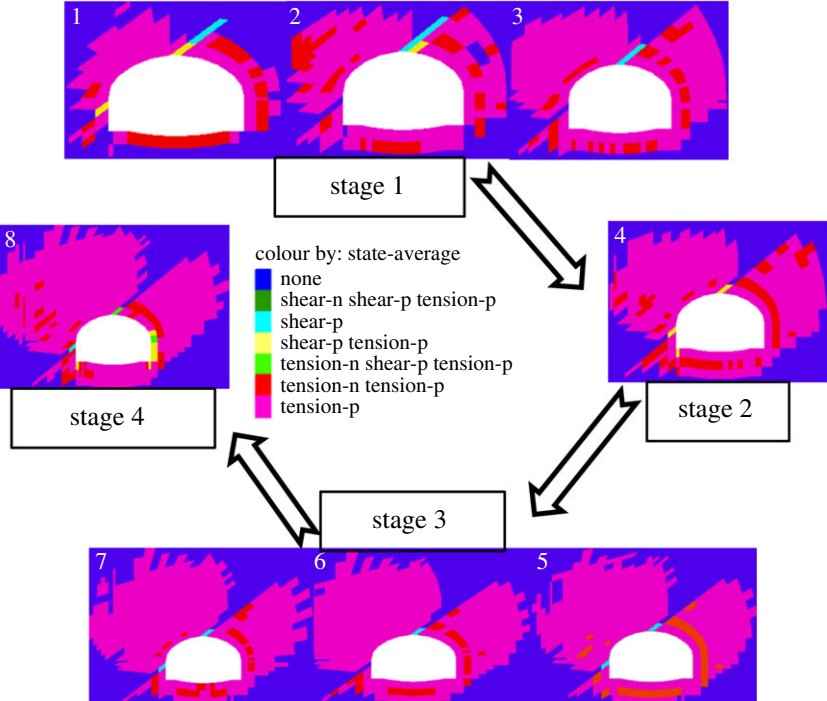

**Figure 17.** Contour of plastic state in case 2.

significantly affected the failure mode. To investigate the joint action of groundwater and the weak interlayer on the failure mode of the tunnel, an extended numerical analysis was employed.

## 5.1. Distribution of pore water pressure

To verify the validity of results for the seepage field obtained by the numerical simulation, the results of pore water pressure in the model test for case 2 were compared with that of the numerical simulation, as shown in figure 18a. The pore water pressure in the interlayer and the surrounding rock was obtained as well by monitoring D1–D8 and J1–J7, respectively, and this is shown in figure 18a,b.

The results show that as excavation progressed, cracks developed and volumetric strain increased in surrounding rock and weak interlayer, causing the redistribution of pore water pressure. Figure 18a shows that the pore water pressure at monitoring points D1–D8 constantly changed. In particular for positions at the vault and the right spandrel, points D1 and D8, the increment in pore water pressure decreased abruptly, which accounted for 42.2% and 32.6% of the total reduced pore water pressure, respectively. The flow mainly converged at the bottom of the arch and the right haunch, points D5 and D7, which accounted for 21.6% and 51% of the total increased pore water pressure, respectively. To prove the validity of these methods, the results of the model test were compared with those of the numerical simulation, and were in good agreement.

Under the action of water, the weak interlayer swelled and became muddy, and had low shear strength [40–43], causing the redistribution of pore water pressure. To investigate this, the corresponding values of increment in pressure were obtained by monitoring points J1–J7, and the relationship between the distribution of pore water pressure and the excavation step is shown in figure 18b.

When the excavation of the first step had been completed, pore water pressure in the weak interlayer was redistributed, and water flow outlet first formed at points J3 and J4, where the weak interlayer intersected with the tunnel. The pore water pressure at these positions decreased significantly, which accounted for 48.7% and 50% of the total reduced pore water pressure, respectively. Subsequently, it remained almost unchanged with excavation. For other measuring points, once the excavation had been completed, the pore water pressure mainly converged at points J1, J5 and J6, and accounted for 32%, 24% and 23% of the total increased pore water pressure, respectively, which means that the flow channel was formed and pore water pressure in the interlayer was evenly distributed.

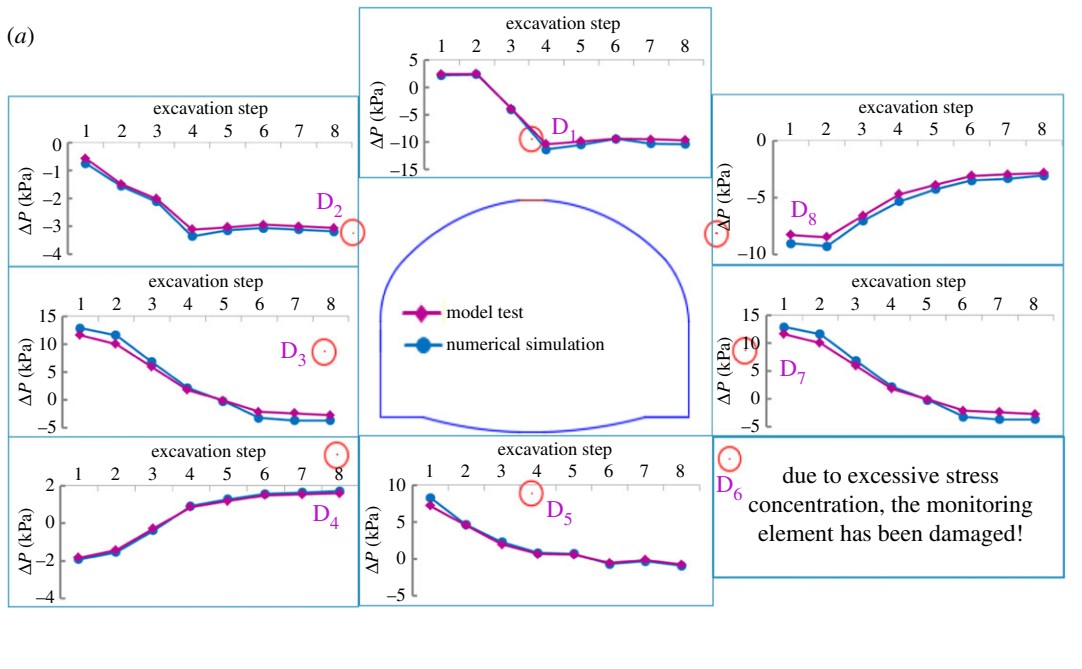

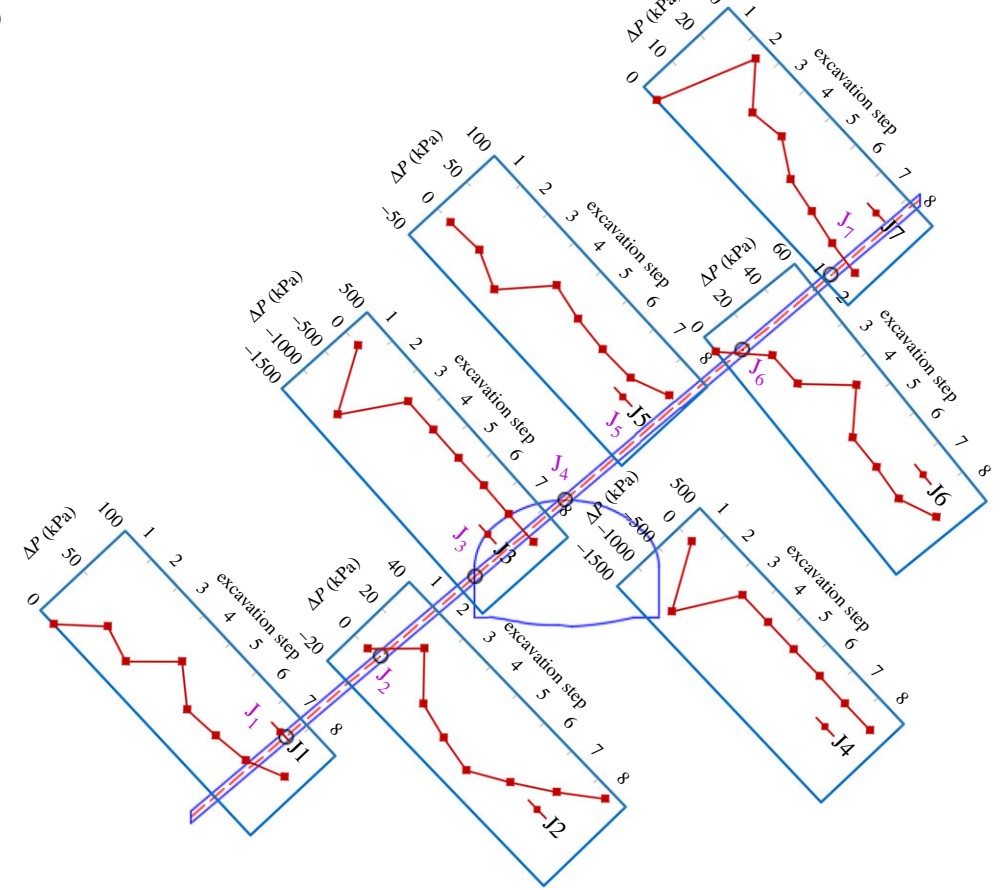

**Figure 18.** Distribution of increment in pore water pressure in case 2. (*a*) Pore water pressure in surrounding rock in simulation and model test and (*b*) in the weak interlayer ($\Delta P$, increment in pore water pressure of surrounding rock).

## 5.2. Changes in law of permeability

Permeability controls the rate of seepage flow in porous and fractured media. Although permeability represents an original geometric property of the porous medium, it can be changed when subjected to

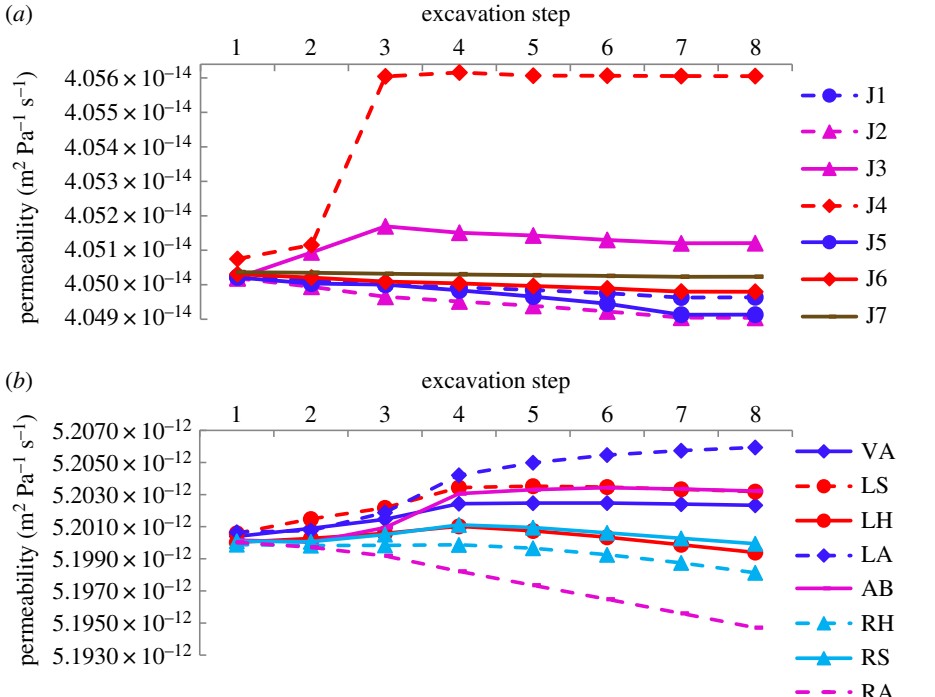

**Figure 19.** Curves of permeability. (*a*) Permeability in the weak interlayer and (*b*) surrounding rock (VA, vault; LS, left spandrel; LH, left haunch; LA, left arch springing; AB, arch bottom; RS, right spandrel; RH, right haunch; RA, right arch springing).

stress variations. Instead of a change in aperture, changes in either void space or grain volume are typical causes of a change in permeability [38]. To investigate dynamic changes in permeability in the weak interlayer and surrounding rocks during excavation, the relationship of permeability with steps of the excavation is shown in figure 19a,b. Figure 19a shows that because points J3 and J4 were the points of intersection of the weak interlayer with surrounding rock, and the left arch haunch and the vault once it had been excavated, owing to the low strength of the interlayer, looseness damage first occurred at these positions. As the surrounding rock pressure decreased, loosening deformation gradually occurred, and promotes the development of micro-fissures and volume strain $\varepsilon_v$, and, correspondingly, permeability increased. Until the completion of the excavation of the third step, it remained nearly unchanged. For other points, permeability remained nearly unchanged, indicating that pore water pressure in these positions exhibited a uniform distribution in the weak interlayer.

According to the results of the model test (shown in figure 11), the released pressure at measuring points D4 (left arch springing) and D2 (left spandrel) was greater than that at the other points, and caused loose deformation and crack propagation. The corresponding permeability increased, as shown in figure 19b. Conversely, because of the part of the pressure distribution at D6 (right arch springing) and D7 (right haunch), the cracks there became compressed, resulting in a reduction in the corresponding permeability.

## 5.3. Evolution of seepage field

The deformation and distribution of the seepage field in the interlayer and surrounding rock during excavation are shown in figure 20. In this image, the thicker the red arrow is, the larger is the specific flow of water. According to characteristics of the distribution of the seepage field, the process of seepage and deformation can be divided into four stages.

In stage 1, at the beginning of excavation, because the weak interlayer was mainly mudstone or shale, initial permeability in it was smaller than that of the surrounding rock (sandstone), which shows that it was waterproof. Therefore, the specific flow of water in the interlayer was almost zero, whereas water around the cavern flowed out of the tunnel face along the radial direction.

In stage 2, in the middle period of excavation, intersections of the weak interlayer with the vault and left haunch were damaged owing to the leftward movement of the hanging wall and downward movement of the footwall, which resulted in a rapid reduction in the specific flow of water. The

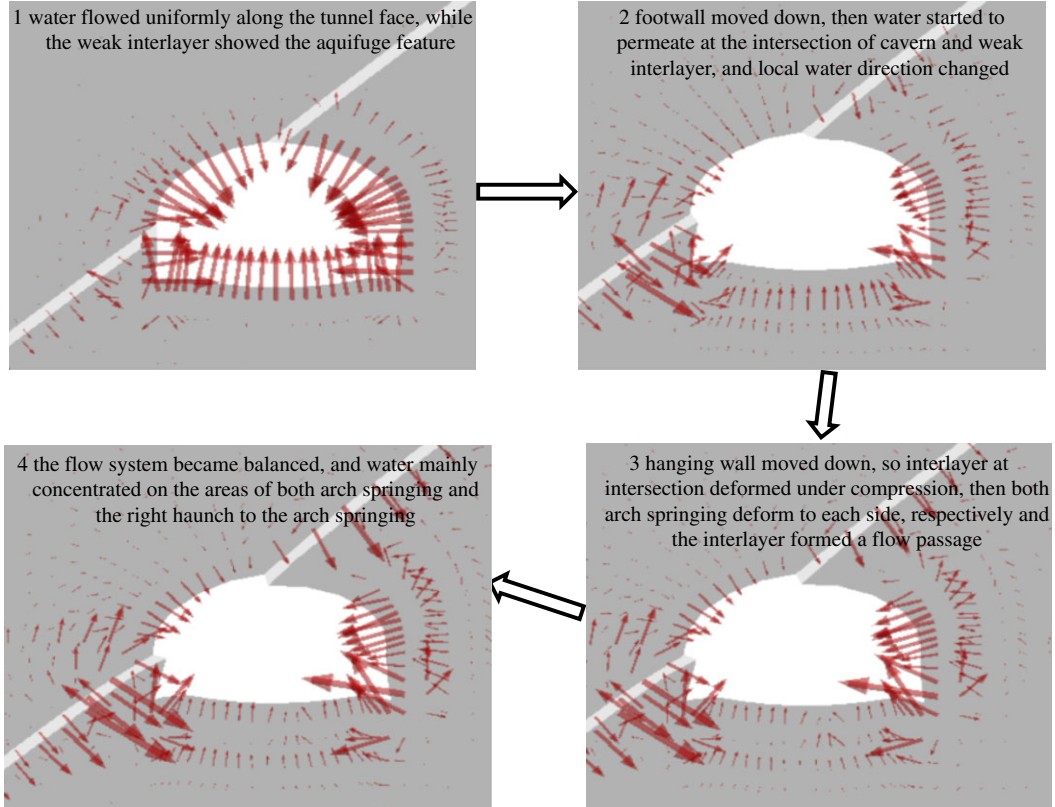

**Figure 20.** Process of seepage field in case 2.

interlayer and both arch springings then deformed due to extrusion, which caused the direction of flow of water to deviate.

In stage 3, leftward deformation occurred at the left arch springing and rightward deformation at the right arch springing because the hanging wall moved downward after it lost support from the footwall. At the same time, there was a trend of diversion in water flow along the inverted arch and the radial direction. With the failure of the weak interlayer, the specific flow of water in interlayer increased while the direction of flow around the cavern became disordered.

In stage 4, when the permeability of the surrounding rocks in water was greater than the ability of the interlayer to resist permeability at a specific time, the interlayer gradually formed a channel of water flow. The deformation in the surrounding rocks and interlayer were then stable, and water in the interlayer had a uniform distribution. The seepage field reached an equilibrium state, and pore water pressure mainly converged at zones of the left arch springing, and that at the right spandrel and the right arch springing.

Therefore, the positions easily damaged were mainly distributed along intersections and the two arch springings, which is consistent with the results of the model test. It was also verified the mechanism that groudwater affecting the failure mode is to reduce the strength and change in permeability by muddying the interlayer.

# 6. Effect of seepage and weak interlayer on failure mechanism

To investigate effects of the coexistence of groundwater and the weak interlayer on failure modes of the surrounding rocks, the differences in the surrounding rock pressure, displacement, and cracks and strain in two cases were analysed.

To understand reasons for why the groundwater induced different failure modes, the authors found the marked difference is the space–time distribution of stress, displacement and cracks. In case 2, the deformation and failure were mainly distributed on the right side of the footwall: namely the right spandrel, haunch and arch springing. On the other hand, distributions of the main cracks ② and ③ were significantly different. For case 2, they were distributed in two arch springings and developed along the arch bottom. For case 1, they were distributed in the left haunch and right arch springing,

and developed towards the deep-surrounding rock on both sides. The third difference was that horizontal displacement on the right side of the footwall was directed towards the right in case 1 but the left in case 2. However, these differences were caused by the groundwater.

The parameters in terms of pore water pressure and permeability, as well as the process of deformation and seepage in the weak interlayer and around the cavern, were simulated and analysed in case 2. Combined with the foregoing analysis, the influential mechanism is explained below.

First, in the initial excavation stage, owing to the unloading effect of excavation, the surrounding rocks lost their initial equilibrium state, and the vault and left haunch as the area of intersection between the interlayer and the surrounding rock became permeable and free face because permeability in the weak interlayer was lower than that in the surrounding rocks, which exhibited waterproof characteristics. Thus, water in the hanging wall flowed out from the vault and the left haunch, whereas water around the cavern flowed out of the tunnel face along the radial direction.

With progression in excavation, the lubrication action of water at the interface of the interlayer and the surrounding rock resulted in bed separation and aggravates of slippage of the interlayer. Then, the hanging wall slipped along the interlayer and the footwall moves downward because it lost support, and crack ① developed. Then, a large amount of water in the hanging wall flowed out rapidly from crack ①, on the one hand, and softened the surrounding rock and muddied the weak interlayer, causing deformation and damage at the vault. On the other hand, part of water flowing into the footwall under the action of gravity damaged the footwall more seriously than the hanging wall. Subsequently, water flowing in the footwall, especially in the right spandrel, aggravated movement towards the bottom-left of the right spandrel and right haunch. Therefore, pressure concentration and rightwards horizontal deformation occurred at the right arch springing, forming crack ②. In addition, because stress transfer beyond the interlayer decreased severely, stress transmitted from the hanging wall and concentrated between tunnel and interlayer [13]. Pressure at the left arch springing thus was released by a large margin under the unloading effect, and the position turned into a loosening zone. The left haunch as the area of intersection, as the interlayer was softened by the flows, increased permeability, and part of the water inflowed into the left springing. Then, the position turned into a loosening zone with rightwards deformation, forming crack ③. Thus, both sides of the arch springing failed to work, which means that the inverted arch began to work.

Moreover, the areas of intersection were weaker than other positions to be damaged, because of which the permeability of the positions gradually increased with excavation. As the interlayer was softened gradually, the numbers of cracks and voids gradually increased and connected in the weak interlayer, increasing its permeability and forming a water flow channel. Finally, flows from the hanging wall entered continuously into the arch bottom along cracks ② and ③, with crack ④ appearing at the bottom of the arch.

According to results obtained from the plastic state in figure 17, the interlayer near the vault was sliding until tunnel excavation had been completed. Thus, pressure transmitted from the hanging wall was concentrated on the right arch springing, forming crack ⑤. Subsequently, the pore water pressure decreased with flow out of crack ⑤. When the hanging wall and footwall met again, a new equilibrium state was formed.

In addition, it should be noted that compared with the results of the seepage field in practice, the results simulated by FlAC3D had discrepancies, especially for the range of surrounding rock with some cracks around the cavern. Referring to the past work [52–55], it is clear that considering the seepage in the fracture can better explain the internal mechanism of the corresponding failure modes from a microscopic point of view. For FlAC3D, the simulation of the generation and propagation of cracks was inadequate. This is also the reason for disturbance along the direction of flow at positions where large deformation occurred. Meanwhile, parameters of weak interlayer in terms of the exposed position, the dip angle and the number are important for the research of failure mechanism. More attention will be given to research in this direction in our future work.

# 7. Conclusion

In this paper, model tests and numerical simulations were carried out to study the effects of the joint action of groundwater and the weak interlayer on the failure mechanism of tunnels. Through a comparison of failure modes between cases 1 and 2, the authors determined that the combined action of water and the weak interlayer affects failure modes of the surrounding rock by changing the properties of the weak interlayer owing to the different space–time distributions of stress and cracks. The following conclusions can be drawn.

The most marked difference in failure modes was in terms of the space–time distributions of stress, displacement and cracks. In case 2, the deformation and damage were mainly distributed in the footwall, and the distributions of main cracks ② and ③ in case 2 were at both sides of the arch springing and developed along the inverted arch in a V shape. In case 1, in contrast with case 2, they were distributed in the left haunch and right arch springing, and developed along the radial direction of the surrounding rock, presenting in a 'reversed V' shape. The third difference was that the horizontal displacement on the right side of the footwall was towards right in case 1 but left in case 2.

The results of the numerical simulation and the model test were in good agreement. And for case 2, the footwall of the tunnel needs to be attended to, and requires timely support.

The joint action of groundwater and weak interlayer on the failure mechanism of the surrounding rock was investigated by numerical analysis in this study. The authors found that they complement each other. The waterproof characteristic of the weak interlayer accelerates the slip in the interlayer and the generation of cracks, where the latter act as the main channels of water flow and guide its direction. The water weakens the strength of the interlayer and surrounding rocks when it seeps into different places, and changes the distribution of stress, displacement and strain in the surrounding rock. As a result, deformations and cracks at different locations form, and lead to different failure modes.

Without considering fracture seepage as the inadequacy in this paper, greater attention will be accorded to this field to further examine the influence of seepage on the failure mechanism of tunnels with weak interlayers.

Data accessibility. Data are available from the Dryad Digital Repository: https://doi.org/10.5061/dryad.sh64br0 [56].
Authors' contributions. J.H. conceived of the method, designed and conducted the model test and numerical analysis, and wrote the paper. H.W. and B.L. reviewed and edited the manuscript. Q.X. and Q.M. helped conduct the model tests. All authors gave final approval for publication.
Competing interests. We declare we have no competing interests.
Funding. This research was funded by the author H.W. based on the funding of National Key R&D Program of China (grant no. 2018YFC1505501), the High Technology and Industrial Technology Development Special of Chongqing Development and Reform Committee (grant no. [2016]1270), the Venture and Innovation Support Program for Chongqing Overseas Returnees (grant no. CX2017125), the Technology Innovation and Application Demonstration of Chongqing Science and Technology Project (grant no. cstc2018jscx-msybX0310) and the Fundamental Research Funds for the Central Universities (grant no. 2018CDPT CG0001/38).
Acknowledgements. The authors thank reviewers for valuable comments and the editors for hard working to improving the manuscript.

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
