## [Reviewer comments · Royal Society Open Science]

Review History

RSOS-190790.R0 (Original submission)

Review form: Reviewer 1 (Hadi Haeri)

Is the manuscript scientifically sound in its present form?

Yes

Are the interpretations and conclusions justified by the results?

Yes

Is the language acceptable?

No

Is it clear how to access all supporting data?

Yes

Do you have any ethical concerns with this paper?

Yes

Have you any concerns about statistical analyses in this paper?

Yes

Recommendation?

Major revision is needed (please make suggestions in comments)

Comments to the Author(s)

1-What is the novelty of the manuscript? It must be explicitly stated in the paper.

2-the structures of the paper is not Suitable.

3-In the reviewer opinion several experimental works have been done and the results are plotted but these tests are not clearly separated and can not be well distinguishable in the text.

4- There are many new studies on the failure modes of surrounding rock. Add some new references.

1-Suggesting a new testing device for determination of tensile strength of concrete

2-Simulating the effect of disc erosion in TBM disc cutters by a semi-infinite DDM

3-Experimental and Numerical Study of Shear Fracture in Brittle Materials with Interference of Initial Double Cracks

4-A review of experimental and numerical investigations about crack propagation

5-Simulating the crack propagation and cracks coalescence underneath TBM disc cutters

6-Simulating the bluntness of TBM Disc Cutters in Rocks using Displacement Discontinuity Method

7-Evaluating the use of mineral pumice in falling zones of internal pressure tunnels (Case study: Water transfer tunnel of Sardasht dam power plant)

Review form: Reviewer 2

Is the manuscript scientifically sound in its present form?

No

Are the interpretations and conclusions justified by the results?

No

Is the language acceptable?

No

Is it clear how to access all supporting data?

Not Applicable

Do you have any ethical concerns with this paper?

No

Have you any concerns about statistical analyses in this paper?

No

Recommendation?

Reject

Comments to the Author(s)

1 Through out the MS, the language should be greatly improved.

2 Lack of many latest reference in this topic. Many related paper can be found in Engineering Geology, Tunnelling and Underground Space Technology

3 The academic contribution of the present study should be mentioned in the last paragraph.

4 Section 2.1 should be presented as Section 2. In addition, more details should be provided, such as the field photo about the weak zone. The excavation method in practice should be mentioned. How consider the method in the physical model.

5 Suggest to put subsection 2.2-2.5 together as Section 3 "Model test"

6 How to consider the interface between two different materials. The contact of the interface would be great influence the measurement results, and also the mechanical behavior of the physical model. Thus, the point should be enhanced.

7 How to build the physical model is also important. Should be presented in details. It would be better to provide the related figures.

8 Add the statements about the monitoring point in Fig. 5. Why and How.

Considering above, the present MS cannot be suggested to publish in the Journal.

Review form: Reviewer 3

Is the manuscript scientifically sound in its present form?

No

Are the interpretations and conclusions justified by the results?

Yes

Is the language acceptable?

No

Is it clear how to access all supporting data?

Not Applicable

Do you have any ethical concerns with this paper?

No

Have you any concerns about statistical analyses in this paper?

No

Recommendation?

Major revision is needed (please make suggestions in comments)

Comments to the Author(s)

The weakening of mechanical properties of the weak interlayer under the action of water is a direct cause of many cavern failure. It is of great significance to carry out in-depth research on this problem. The overall data in this article is informative, and the results of model tests and

numerical simulations have formed favorable support for the conclusions. However, there are some problems, some are major, that need further improvement. In terms of language, the authors are suggested to find professionals to polish.

In the title, "Mode tests" should be "model tests".

There are some sayings that should be changed. For example, in the abstract, "Model tests results were verified using a numerical analysis model". Only the numerical model requires a physical model to verify. The physical model itself is not verified by a numerical model. However, the numerical model can explain the physical model on mechanisms of the failure process.

The "seepage field" itself is not a seepage parameter. The boundary condition can be used as an input parameter for sensitivity analysis. The content involved in this part needs to be modified, and the basic concept is not clear.

Page 3, line10, "[25-30] investigated the stability of the tunnel face under see page flow conditions, using improved methods." This sentence has no subject.

Similar materials determines the effectiveness of the physical test. The paper should provide a more detailed description of the selection of similar materials. Authors should explain how the similarity of the water properties of the filler (such as water softening) be considered.

The simulation of the in-situ in-situ stress also needs to be carefully explained. Figure 5 shows that the device can only be loaded in the vertical direction, and how the horizontal in-situ stress is simulated should be explained.

The boundary conditions of the seepage are not clearly stated and need to be detailed and illustrated with pictures. The direction of the front and back of the model is in contact with the atmosphere, which will inevitably affect the water pressure distribution during the test. Is there any sealing of the test object during the test? It should be explained in detail.

In Table 1, the permeability coefficient of the weak interlayer is two orders of magnitude lower than that of the surrounding rock. Please confirm that the permeability of the weak interlayer is generally considered to be large. The compressive strength should be uniaxial compressive strength.

The section of the surrounding rock cavern is usually analyzed as a plane strain problem, and the model test is much closer to a plane stress problem. The rationality of this approach should be illustrated. If it is reasonable, the similarity criteria need to be modified accordingly.

The failure mode is only analyzed by surface cracks. The question is whether the internal failure mode can be further tested.

Both displacement field and seepage field were not compared quantitatively between numerical and physical tests. The lack of quantitative comparison of experimental data and numerical simulation results is of great significance for the final reception of the article.

The results of numerical simulation of seepage is very strange. There are multiple different flow directions at almost the same point near the lower left corner of the chamber. For different failure modes, fracture seepage analysis is important. Authors can refer to the following works for clarification.

Seepage flow with free surface in fracture networks. *Water Resources Research*, 2013, 49(1):176-186

A numerical procedure for transient free surface seepage through fracture networks. *Journal of Hydrology*, 2014, 519: 881–891

A numerical analysis of permeability evolution in rocks with multiple fractures. *Transport in Porous Media*, 2015, 108(2):289–311

Influences of connectivity and conductivity on nonlinear flow behaviours through three-dimension discrete fracture networks. *Computers and Geotechnics*, 107(2019) 128–141

Review form: Reviewer 4

Is the manuscript scientifically sound in its present form?

No

Are the interpretations and conclusions justified by the results?

Yes

Is the language acceptable?

No

Is it clear how to access all supporting data?

Yes

Do you have any ethical concerns with this paper?

No

Have you any concerns about statistical analyses in this paper?

No

Recommendation?

Major revision is needed (please make suggestions in comments)

Comments to the Author(s)

Comments to the Author

Studying the effect of seepage and weak interlayer on the failure modes of surrounding rocks mass is of significance for rock engineering, especially for tunnel engineering. The article presents some interesting test results both with numerical model and physical model. After going through the article, I think that this paper has a lot of potential, but it needs thorough editing before it is ready for publication. The main suggestions are as follows.

(1) The structure of “Abstract” section is confusing. I would recommend that the authors look at the structure and balance of material in papers published in the journal of “Royal Society Open Science” and use this process to restructure your own paper and provide more information on the results.

(2) The language needs polishing to get rid of the grammatical errors, e.g. for section 1, in line 1 of paragraph 2, a subject is missing before “has”, in line 1 of paragraph 3, “except” is suggested to be replaced by “in addition to”, in line 3 of paragraph 4, “It” should be smaller case and so on.

- (3) In the first line of section 2.3, "According to the engineering background and the similarity theory," and "According to the similarity theory," they are repetitive in expression.
- (4) The background information part needs to be shortened and the physical mechanisms lying behind the experimental phenomena should be paid more attention to. It is more significant to dig out the reasons to explain the results.
- (5) The literature review needs to be improved, some related newest investigations are suggested to be added.
- (6) The results of model tests and numerical simulations should be compared to verify each other.
- (7) The boundary conditions are not clear for both mechanical and seepage analysis.
- (8) After the numerical simulation is verified by the physical model test, effects of the relevant parameters, such as the exposed position, the dip angle, and the number of the weak interlayers, should be further explored. This point on which the article needs to be strengthened.

Decision letter (RSOS-190790.R0)

18-Jun-2019

Dear Dr Hu,

The editors assigned to your paper ("The effect of seepage and weak interlayer on the failure modes of surrounding rock - Mode tests and numerical analysis") have now received comments from reviewers. We would like you to revise your paper in accordance with the referee and Associate Editor suggestions which can be found below (not including confidential reports to the Editor). Please note this decision does not guarantee eventual acceptance.

Please submit a copy of your revised paper before 11-Jul-2019. Please note that the revision deadline will expire at 00.00am on this date. If we do not hear from you within this time then it will be assumed that the paper has been withdrawn. In exceptional circumstances, extensions may be possible if agreed with the Editorial Office in advance. We do not allow multiple rounds of revision so we urge you to make every effort to fully address all of the comments at this stage. If deemed necessary by the Editors, your manuscript will be sent back to one or more of the original reviewers for assessment. If the original reviewers are not available, we may invite new reviewers.

- Data accessibility

If you wish to submit your supporting data or code to Dryad (<http://datadryad.org/>), or modify your current submission to dryad, please use the following link:
<http://datadryad.org/submit?journalID=RSOS&manu=RSOS-190790>

- Competing interests

- Authors' contributions

- Acknowledgements

- Funding statement

on behalf of R. Kerry Rowe (Subject Editor)
 openscience@royalsociety.org

Comments to Author:

Reviewers' Comments to Author:
 Reviewer: 1

Comments to the Author(s)

- 1-What is the novelty of the manuscript? It must be explicitly stated in the paper.
- 2-the structures of the paper is not Suitable.
- 3-In the reviewer opinion several experimental works have been done and the results are plotted but these tests are not clearly separated and can not be well distinguishable in the text.
- 4- There are many new studies on the failure modes of surrounding rock. Add some new references.

- 1-Suggesting a new testing device for determination of tensile strength of concrete
- 2-Simulating the effect of disc erosion in TBM disc cutters by a semi-infinite DDM
- 3-Experimental and Numerical Study of Shear Fracture in Brittle Materials with Interference of Initial Double Cracks
- 4-A review of experimental and numerical investigations about crack propagation
- 5-Simulating the crack propagation and cracks coalescence underneath TBM disc cutters
- 6-Simulating the bluntness of TBM Disc Cutters in Rocks using Displacement Discontinuity Method
- 7-Evaluating the use of mineral pumice in falling zones of internal pressure tunnels (Case study: Water transfer tunnel of Sardasht dam power plant)

Reviewer: 2

Comments to the Author(s)

- 1 Through out the MS, the language should be greatly improved.
- 2 Lack of many latest reference in this topic. Many related paper can be found in Engineering Geology, Tunnelling and Underground Space Technology
- 3 The academic contribution of the present study should be mentioned in the last paragraph.

4 Section 2.1 should be presented as Section 2. In addition, more details should be provided, such as the field photo about the weak zone. The excavation method in practice should be mentioned. How consider the method in the physical model.

5 Suggest to put subsection 2.2-2.5 together as Section 3 "Model test"

6 How to consider the interface between two different materials. The contact of the interface would be great influence the measurement results, and also the mechanical behavior of the physical model. Thus, the point should be enhanced.

7 How to build the physical model is also important. Should be presented in details. It would be better to provide the related figures.

8 Add the statements about the monitoring point in Fig. 5. Why and How.

Considering above, the present MS cannot be suggested to publish in the Journal.

Reviewer: 3

Comments to the Author(s)

The weakening of mechanical properties of the weak interlayer under the action of water is a direct cause of many cavern failure. It is of great significance to carry out in-depth research on this problem. The overall data in this article is informative, and the results of model tests and numerical simulations have formed favorable support for the conclusions. However, there are some problems, some are major, that need further improvement. In terms of language, the authors are suggested to find professionals to polish.

In the title, "Mode tests" should be "model tests".

There are some sayings that should be changed. For example, in the abstract, "Model tests results were verified using a numerical analysis model". Only the numerical model requires a physical model to verify. The physical model itself is not verified by a numerical model. However, the numerical model can explain the physical model on mechanisms of the failure process.

The "seepage field" itself is not a seepage parameter. The boundary condition can be used as an input parameter for sensitivity analysis. The content involved in this part needs to be modified, and the basic concept is not clear.4.

Page 3, line10, "[25-30] investigated the stability of the tunnel face under see page flow conditions, using improved methods." This sentence has no subject.

Similar materials determines the effectiveness of the physical test. The paper should provide a more detailed description of the selection of similar materials. Authors should explain how the similarity of the water properties of the filler (such as water softening) be considered.

The simulation of the in-situ in-situ stress also needs to be carefully explained. Figure 5 shows that the device can only be loaded in the vertical direction, and how the horizontal in-situ stress is simulated should be explained.

The boundary conditions of the seepage are not clearly stated and need to be detailed and illustrated with pictures. The direction of the front and back of the model is in contact with the atmosphere, which will inevitably affect the water pressure distribution during the test. Is there any sealing of the test object during the test? It should be explained in detail.

In Table 1, the permeability coefficient of the weak interlayer is two orders of magnitude lower

than that of the surrounding rock. Please confirm that the permeability of the weak interlayer is generally considered to be large. The compressive strength should be uniaxial compressive strength.

The section of the surrounding rock cavern is usually analyzed as a plane strain problem, and the model test is much closer to a plane stress problem. The rationality of this approach should be illustrated. If it is reasonable, the similarity criteria need to be modified accordingly.

The failure mode is only analyzed by surface cracks. The question is whether the internal failure mode can be further tested.

Both displacement field and seepage field were not compared quantitatively between numerical and physical tests. The lack of quantitative comparison of experimental data and numerical simulation results is of great significance for the final reception of the article.

The results of numerical simulation of seepage is very strange. There are multiple different flow directions at almost the same point near the lower left corner of the chamber. For different failure modes, fracture seepage analysis is important. Authors can refer to the following works for clarification.

Seepage flow with free surface in fracture networks. *Water Resources Research*, 2013, 49(1):176-186

A numerical procedure for transient free surface seepage through fracture networks. *Journal of Hydrology*, 2014, 519: 881-891

A numerical analysis of permeability evolution in rocks with multiple fractures. *Transport in Porous Media*, 2015, 108(2):289-311

Influences of connectivity and conductivity on nonlinear flow behaviours through three-dimension discrete fracture networks. *Computers and Geotechnics*, 107(2019) 128-141

Reviewer: 4

Comments to the Author(s)

Studying the effect of seepage and weak interlayer on the failure modes of surrounding rocks mass is of significance for rock engineering, especially for tunnel engineering. The article presents some interesting test results both with numerical model and physical model. After going through the article, I think that this paper has a lot of potential, but it needs thorough editing before it is ready for publication. The main suggestions are as follows.

(1) The structure of "Abstract" section is confusing. I would recommend that the authors look at the structure and balance of material in papers published in the journal of "Royal Society Open Science" and use this process to restructure your own paper and provide more information on the results.

(2) The language needs polishing to get rid of the grammatical errors, e.g. for section 1, in line 1 of paragraph 2, a subject is missing before "has", in line 1 of paragraph 3, "except" is suggested to be replaced by "in addition to", in line 3 of paragraph 4, "It" should be smaller case and so on.

(3) In the first line of section 2.3, "According to the engineering background and the similarity theory," and "According to the similarity theory," they are in repetitive in expression.

(4) The background information part needs to be shortened and the physical mechanisms lying

behind the experimental phenomena should be paid more attentions to. It is more significant to dig out the reasons to explain the results.

(5) The literature review need to be improved, some related newest investigations are suggested to be added.

(6) The results of model tests and numerical simulations should be compared to verify each other.

(7) The boundary conditions are not clear for both mechanical and seepage analysis.

(8) After the numerical simulation is verified by the physical model test, Effects of the relevant parameters, such as the exposed position, the dip angle, and the number of the weak interlayers, should be further explored. This point on which the article needs to be strengthened.

Comments to the Authors from the Editorial Office:

For more information about language editing services endorsed by the Royal Society, please follow the link below:

<https://royalsociety.org/journals/authors/language-polishing/>

Author's Response to Decision Letter for (RSOS-190790.R0)

See Appendix A.

RSOS-190790.R1 (Revision)

Review form: Reviewer 2

Is the manuscript scientifically sound in its present form?

Yes

Are the interpretations and conclusions justified by the results?

Yes

Is the language acceptable?

Yes

Do you have any ethical concerns with this paper?

No

Have you any concerns about statistical analyses in this paper?

No

Recommendation?

Accept as is

Comments to the Author(s)

No.

Review form: Reviewer 3

Is the manuscript scientifically sound in its present form?

Yes

Are the interpretations and conclusions justified by the results?

Yes

Is the language acceptable?

Yes

Do you have any ethical concerns with this paper?

No

Have you any concerns about statistical analyses in this paper?

No

Recommendation?

Accept with minor revision (please list in comments)

Comments to the Author(s)

The revised version seems read for publication. The authors have addressed all my comments and suggestions satisfactorily and I can now recommend the paper for publication. It is noticed that, in the references [52-55], Jiang QJ should be modified as Jiang QH.

Review form: Reviewer 4

Is the manuscript scientifically sound in its present form?

Yes

Are the interpretations and conclusions justified by the results?

Yes

Is the language acceptable?

Yes

Do you have any ethical concerns with this paper?

No

Have you any concerns about statistical analyses in this paper?

No

Recommendation?

Accept as is

Comments to the Author(s)

My problems are well cleared. This article can be accepted as it is.

Decision letter (RSOS-190790.R1)

14-Aug-2019

Dear Dr Wen,

I am pleased to inform you that your manuscript entitled "The effect of seepage and weak interlayer on the failure modes of surrounding rock - Model tests and numerical analysis" is now accepted for publication in Royal Society Open Science.

on behalf of Prof R. Kerry Rowe (Subject Editor)
openscience@royalsociety.org

Reviewer comments to Author:

Reviewer: 4

Comments to the Author(s)

My problems are well cleared. This article can be accepted as it is.

Reviewer: 2

Comments to the Author(s)

No.

Reviewer: 3

Comments to the Author(s)

The revised version seems read for publication. The authors have addressed all my comments and suggestions satisfactorily and I can now recommend the paper for publication. It is noticed that, in the references [52-55], Jiang QJ should be modified as Jiang QH.

Appendix A

Author Response to Reviewers' Comments for RSOS-190790

List of response

Dear Editor and Reviewers:

Firstly, the authors greatly appreciate your letter and the reviewers' comments concerning the manuscript entitled "The effect of seepage and weak interlayer on the failure modes of surrounding rock - Model tests and numerical analysis". Those comments are professional and valuable for revising and improving the paper, as well as the important guiding significance to my researches. According to these comments, we amended the relevant parts in our manuscript in red, and all comments were addressed. We hope the revision could meet with approval. Some of paragraphs, figures and Tables in the paper have been renumbered. The main corrections in the paper and the responds to the reviewer's comments are as follows:

Responds to the reviewer's comments:

EDITORIAL OFFICE'S COMMENTS FOR THE AUTHOR:

For more information about language editing services endorsed by the Royal Society, please follow the link below:

<https://royalsociety.org/journals/authors/language-polishing/>

A: The authors are appreciate your hard work and encouragements! We have chosen language editing services endorsed by the Royal Society to polish the language of the paper. The polishing certificate is attached as follows.

EDITORIAL CERTIFICATE

This document certifies that the manuscript below was edited for correct English language usage, grammar, punctuation and spelling by qualified native English speaking editors at Charlesworth Author Services.

Paper Title:

The effect of seepage and weak interlayer on the failure modes of surrounding rock - Model tests and numerical analysis

Author:

jing Hu

Date certificate issued:

July 12, 2019

cwauthors.com

REVIEWER'S COMMENTS FOR THE AUTHOR:

Reviewer: 1

Comments to the Author

A: First of all, thanks for reviewer's encouragement and those comments, and those comments are constructive and positive for revising and improving the paper.

1-What is the novelty of the manuscript? It must be explicitly stated in the paper.

A: I'm sorry for not explaining the novelty of the manuscript clearly. Based on your suggestions, I have sorted out the paper, especially for the content and logic of abstract and introduction have been adjusted and revised. In this paper, the main research contents are the mechanical and failure modes of the tunnel under the joint action of water and a weak interlayer.

A lot of researches on failure modes in two undesirable geological conditions have been respectively by the predecessors, whereas research on the effect of the coexistence of seepage and a weak interlayer has dealt mainly with slope stability, and scant work has considered this scenario in tunnels. What's more, the coexistence of weak interlayers and groundwater are common in tunnels, and will complicate the geological conditions, causing the difference of failure modes. Therefore, it is important to understand the mechanical and failure modes of the tunnel under the joint action of water and a weak interlayer to design the requisite support and ensure safe excavation.

The details of the revision information, please, see in abstract and paragraph 4 of section 1.

2-the structures of the paper is not suitable.

A: I agree with the comment of the reviewer very much. By referring to literature published in "Royal Society Open Science", I have sorted out the full text carefully and re-adjust the structure and logic of this paper, the ideas of the revised paper are as follows:

Firstly, model tests and numerical simulations were conducted in this study based on two cases to investigate the difference of failure modes of surrounding rock. To be specific, the characteristic of surrounding rocks pressure and progressive failure model for two cases are compared and analysed during excavation. Then, the results of experiment and numerical analysis are compared qualitatively and quantitatively, which proves the reliability of the numerical simulation method. Finally, in order to investigate the mechanism causing the difference of failure modes, variation characteristics of seepage field are analyzed, and the effect of seepage and weak interlayer on failure mechanism are discussed. The detailed revision information is shown in catalogue and corresponding chapters.

Thanks for your helping comment and time again!

3-In the reviewer opinion several experimental works have been done and the results are plotted but these tests are not clearly separated and can not be well distinguishable in the text.

A: Thank you for your recognition of my experimental works. According the constructive comment, we have made great changes to the paper, in the process of adjusting structure and logic of this paper, the corresponding test results were also sorted out. The detail information of revision of test results is shown in section 4 of the paper and section 7.

4- There are many new studies on the failure modes of surrounding rock. Add some new references.

A: According to reviewer's comment, I have read some the latest relevant literatures on the failure modes of surrounding rock, it's helping for my study. And they are added in paragraph 1 of introduction and list in reference list (1-7).

Thank you very much again!

Reviewer Comment 2

Comments to the Author

A: First of all, it's appreciated for reviewer's hard work and valuable comments for revising and improving the

paper.

1 Through out the MS, the language should be greatly improved.

A: Thank you for your comment, we have chosen language editing services endorsed by the Royal Society to polish the language of the paper.

2 Lack of many latest reference in this topic. Many related paper can be found in Engineering Geology, Tunnelling and Underground Space Technology

A: Thank you for your helping comment, I agree with the comment of the reviewer very much. I have read some the latest relevant literatures on failure modes of surrounding rock and seepage in fissures, it has great helping for my research. And they are added in paragraph 1 of introduction and the last paragraph of section 6, and list in reference list (1-7) and (52-55).

3 The academic contribution of the present study should be mentioned in the last paragraph.

A: Thanks for your comment, the logic of the introduction has been readjusted, and the detailed information of academic contributions of the present study is shown in paragraph 4 of the introduction.

4 Section 2.1 should be presented as Section 2. In addition, more details should be provided, such as the field photo about the weak zone. The excavation method in practice should be mentioned. How consider the method in the physical model.

A : Thanks for your comment, I agree with it very much!

① By sorting out the structure of the article, after supplemented some details, such as the field photo about the weak zone, Section 2.1 has been as a separate section 2. Please allow me to make an explanation that considering the another reviewer's suggestion to shorten the engineering background, so, in this paper, I add the typical field photo about the weak zone, and briefly introduce the background of the project. The add information are shown in lines 6-7 of the first paragraph of section 2 and Figure 2.

② In addition, because the tunnel is a small-span, IV-grade surrounding rock, so the full-face excavation method is adopted in practice. In order to be consistent with practice, A device was designed specially for the simulation of tunnel excavation. please see in Figure 5 and paragraph 2 of subsection 3.2.

5 Suggest to put subsection 2.2-2.5 together as Section 3 "Model test"

A: Thanks for your helpful comment, By sorting out the articles, we have made great changes to the structure of the paper. So, Previous subsections 2.2-2.5 superadd the introduction of numerical simulation methods are put together as section 3 materials and methods. The detail information of revised are in section 3.

6 How to consider the interface between two different materials. The contact of the interface would be great influence the measurement results, and also the mechanical behavior of the physical model. Thus, the point should be enhanced.

A: Thanks for the reviewers'professional comments, accurate treatment of interface between two different materials is indeed a difficult problem in model test, here I refer to the treatment method of model tests with similar engineering background. Considering the importance of interface, a mixture of sand and clay was used as spacer between the weak interlayer and the surrounding rock, and the point is enhanced in Lines 3-4 of paragraph 6 of subsection 3.3.

7 How to build the physical model is also important. Should be presented in details. It would be better to provide the related figures.

A: Thanks for the reviewers'professional comments, I am sorry that the description of the test process is not exhaustive enough. The building process of the physical model are added in paragraph 5-11 of subsection 3.3, and corresponding figures are Figure 6(a)and 9(a-f).

8 Add the statements about the monitoring point in Fig. 5. Why and How.

A: Thanks for the reviewers'professional comments, I am sorry that the description of monitoring point are not clearly. Figure 6(c-d) shows the lateral and longitudinal distribution of monitoring points, Figure 6(a) and 9(a) shows the specific buried position of monitoring elements in model test.

To be Specifically, to reduce the disturbance of excavation, the buried depth of the earth pressure cell was 10 cm in the radial direction and 15 cm in the longitudinal direction from the cavern. The buried position of the seepage pressure meters was 10 cm in the radial direction and 25 cm in the longitudinal direction from the cavern.

Meanwhile, the points D1-D8 in all the figures are the monitoring points located at the vault, left spandrel, left haunch, left arch springing, arch bottom, right spandrel, right haunch and right arch springing respectively. Similarly, points J1-J7 are the monitoring points located at the weak interlayers.

After each step of excavation is completed and the system is in a static state stabilized, a DH3816 data collection box was used to collect monitored data. Details of the revision information are shown in paragraph 3 of subsection 3.2 and the last sentence of the 10 paragraphs in subsection 3.3.

Considering above, the present MS cannot be suggested to publish in the Journal.

Thank you again for your time, according to your comments, I have revised the text based on your comments one by one, and I have to say that your suggestion has given me more in-depth thinking about the experiment. Based on the above, I have made great changes to the structure, logic and content of the full text, and the reasons to explain results are further dug out as well. If possible, please take this paper into account, thank you very much!

Reviewer: 3

Comments to the Author

The weakening of mechanical properties of the weak interlayer under the action of water is a direct cause of many cavern failure. It is of great significance to carry out in-depth research on this problem. The overall data in this article is informative, and the results of model tests and numerical simulations have formed favorable support for the conclusions. However, there are some problems, some are major, that need further improvement. In terms of language, the authors are suggested to find professionals to polish.

Firstly, thanks for reviewer's encouragement and those constructive comments

(1) **In the title, "Mode tests" should be "model tests" .**

A: I'm sorry for the negligence, and thank you for your remind and your time again, I have revised it in the title. Please see the title in red.

(2) **There are some sayings that should be changed. For example, in the abstract, "Model tests results were verified using a numerical analysis model". Only the numerical model requires a physical model to verify. The physical model itself is not verified by a numerical model. However, the numerical model can explain the physical model on mechanisms of the failure process.**

A: First of all, I agree with the reviewer's comments very much. I'm sorry for the inappropriate presentation of language, all the similar explanations have been revised in full text. The detail information of revision are in lines 2-3 of paragraph 4 of Subsection 4.2, and the last three lines of paragraph 2 in subsection 5.1.

(3) **The "seepage field" itself is not a seepage parameter. The boundary condition can be used as an input parameter for sensitivity analysis. The content involved in this part needs to be modified, and the basic concept is not clear.**

A: Thank you for your professional and careful review.

① The concepts of seepage field and seepage parameter are confirmed by referring relevant literature. So, the structure, subsection and content of previous section 4.3 are adjusted and revised. The detail of revised information are shown in section 5.

② The boundary condition of model test are as follow. From Figure 1 and 2 above, you can see that X-direction constraint is applied to both left and right boundaries, Y-direction constraint is applied to front and rear boundaries, vertical constraints in Z- direction applied to the bottom of the model, the surface of the model is set as a free surface. It is worth noting that, before adding water, the front and rear transparent Plexiglas plates are sealed, as seen in Figure 2 above. Besides all the boundary conditions of numerical model are the same as that of the physical model tests. The detail of revised information are shown in lines 8-12 of the first paragraph of section 3.4.

Figure 1 Model box

Figure 2 Sealing schematic diagram of model box

(4) Page 3, line10, "[25-30] investigated the stability of the tunnel face under see page flow conditions, using improved methods." This sentence has no subject.

A: Thanks for your meticulous review! The language of the full text has been polished by professional polishing agencies. The subject has been added in lines 10-11 of paragraph 3 of section 1.

(5) Similar materials determines the effectiveness of the physical test. The paper should provide a more detailed description of the selection of similar materials. Authors should explain how the similarity of the water properties of the filler (such as water softening) be considered.

A: First of all, thanks for your professional comments I agree with you very much.

① A more detailed description of the selection of similar materials is as follow and added in Paragraph 2 of subsection 3.1:

The similarity material involved two kinds of mixture ratio tests: surrounding rock and weak interlayer. Based on the mechanical parameters (shown in Table 1) and the general law of similarity (shown in Table 2), the physical parameters of the surrounding rocks and weak interlayer in the model test were deduced as shown in Table 3. The similarity mixture selected for the surrounding rock consisted of sand, white cement, barite powder, talcum powder, silicone oil, petrolatum and gypsum. This has been shown to be appropriate for underground fluid–solid coupling model tests by Li [45]. For the weak interlayer, the similarity mixture consisted of clay, river sand, talc powder and silicone oil. It has been shown to be appropriate for the seepage analysis of the weak interlayer [46].

② For the similarity of the water properties of the filler, it is mainly considered from the perspective of permeability coefficient. According to similarity theory, the relationship between similar permeability coefficient, similar time and similar constant is shown in equation (1) and in Table 3 of paper. The specific values of original rock and model tests are located at Table 1 and Table 4 in red.

$$C_k = C_t = \sqrt{C_l} \quad (1)$$

(6) The simulation of the in-situ stress also needs to be carefully explained. Figure 5 shows that the device can only be loaded in the vertical direction, and how the horizontal in-situ stress is simulated should be explained.

A: As shown in Figure 3 below, firstly, overburden depths of the tunnel can be achieved by loading on the top of model ground, whose value is calculated by the equation (2). Then, as seen in Figure 1 above, the constraint along the X-direction was applied to both the left and right boundaries, and that along the Y-direction was applied to the front and rear boundaries. Vertical constraints along the Z-direction were applied to the bottom

of the model, the surface of the model was set as a free surface. Therefore, the reaction force of transparent Plexiglas plate restrained in front and back and steel frames restrained in left and right of the model box to the model is used to simulate the confining pressure of model.

$$H = \left(\frac{P}{\gamma} + 0.608\right)C_l \approx 1.837e-3P + 27.36, \quad (2)$$

Figure 3 Loading and Monitoring System

(7) The boundary conditions of the seepage are not clearly stated and need to be detailed and illustrated with pictures. The direction of the front and back of the model is in contact with the atmosphere, which will inevitably affect the water pressure distribution during the test. Is there any sealing of the test object during the test? It should be explained in detail.

A: I'm sorry that the boundary conditions of water and the excavation process of mode test are not explained clearly, which may have caused some misunderstandings.

① As shown in Figure 1, except for the surface of the model was set as a free surface, all the boundary conditions of numerical model were normal constrain.

② To saturate the model with water, glass glue was used to seal the gap in the Plexiglas between the surrounding rock and the entrance to the tunnel, as shown in Figure 2 above. So, the direction of the front and back of the model is completely cut off with the atmosphere. The detail of the revised information is shown in lines 1-2 of paragraph 9 of subsection 3.3.

(8) In Table 1, the permeability coefficient of the weak interlayer is two orders of magnitude lower than that of the surrounding rock. Please confirm that the permeability of the weak interlayer is generally considered to be large. The compressive strength should be uniaxial compressive strength.

A: Thanks for your careful review and professional comments!

① I'm sorry for the neglect, indeed, unconfined compression tests is used to measure the uniaxial strength, as shown in Figure 4 above. The revision details is shown in Table1 and 4.

② I quite approve of the idea that permeability of the weak interlayer is generally considered to be large, however, the weak interlayer of the engineering based on are mainly composed of mudstone, and shale and coal series. Based on the selected similar materials and many groups of indoor permeability test, it is found that the permeability coefficient of the weak interlayer is less than that of the weak interlayer in two orders of magnitude. So, in this model test and numerical simulation, we study the case that the permeability of weak interlayer is less than that of surrounding rock. And the case that the permeability of weak interlayer is larger than that of surrounding rock will be carried out from the point of view of different filler of weak interlayer, in the future research.

Figure 4 Test of uniaxial compressive strength

(9) The section of the surrounding rock cavern is usually analyzed as a plane strain problem, and the model test is much closer to a plane stress problem. The rationality of this approach should be illustrated. If it is reasonable, the similarity criteria need to be modified accordingly.

A: Distinguished reviewer, I feel confused that if there are some things misunderstanding about the model box. Such as, because the Plexiglas plate in both side of front and back are transparent, shown in Figure 2 above, so it looks like there are no constraints on the both side of front and back of the model box.

If so, I'm sorry for that my statement of boundary of model test is not exhaustive enough. According to Zhang(Zhang, 2009), If the weak interlayer is parallel to the intermediate principal stress and/or the intermediate stress is equal to the minimum principal stress, the three-dimensional model can be reduced to a two-dimensional, and analyzed as plane strain problem.

As seen in Figure1above, except for the surface of the model was set as a free surface, all the boundary conditions of model test are normal constrain. Meanwhile, composition material of model box has enough stiffness to resist deformation. So, I think this is simplified to a plane strain problem.

(10) The failure mode is only analyzed by surface cracks. The question is whether the internal failure mode can be further tested.

A: Thank you for your helpful comments. Through sorting out the paper and adjusting the structure of the articles, the contents of the articles are further dug out. The detail information of revision is shown in section 7.

(11) Both displacement field and seepage field were not compared quantitatively between numerical and physical tests. The lack of quantitative comparison of experimental data and numerical simulation results is of great significance for the final reception of the article.

A: Thank you for your helpful comments. The results of displacement field and seepage field in numerical and physical tests are compared quantitatively. The detail information of revision is shown in Figure 14 and paragraph 4 of section 4.2, and Figure 18(a) and line 6-8 of paragraph 2 of section 5.1. The results of the model test were in good agreement with those of the numerical simulation.

(12) The results of numerical simulation of seepage is very strange. There are multiple different flow directions at almost the same point near the lower left corner of the chamber. For different failure modes, fracture seepage analysis is important. Authors can refer to the following works for clarification.

A: Thank you for your professional comment, they are very helpful for me to improve my paper. It should be noted that because for FIAC3D, the simulation of the generation and propagation of cracks was inadequate, especially for the range of surrounding rock with some cracks around the cavern. This is the reason for disturbance

along the direction of flow at positions where large deformation occurred, and resulting in that it seems multiple different flow directions at almost the same point near the lower left corner of the chamber. Referring to the past work[1-4] it is clear that considering the seepage in the fracture can better explain the internal mechanism of the corresponding failure modes from a microscopic point of view. This direction will be considered by the author in future work. Details of the revision are show in the last paragraph of section 6.

Reviewer: 4

Comments to the Author

Studying the effect of seepage and weak interlayer on the failure modes of surrounding rocks mass is of significance for rock engineering, especially for tunnel engineering. The article presents some interesting test results both with numerical model and physical model. After going through the article, I think that this paper has a lot of potential, but it needs thorough editing before it is ready for publication. The main suggestions are as follows.

First of all, thanks for reviewer's encouragement and those comments are constructive and positive for revising and improving the paper.

(1) The structure of "Abstract" section is confusing. I would recommend that the authors look at the structure and balance of material in papers published in the journal of "Royal Society Open Science" and use this process to restructure your own paper and provide more information on the results.

A: I agree with reviewer's comment very much. According the professional comment, we have made great changes to the article, especially the adjustment of the structure of the article to make the logic of the article clearer. Therefore, the abstract has been revised and the conclusion has been drawn in depth. The detail information of revision is shown in abstract and conclusions.

(2) The language needs polishing to get rid of the grammatical errors, e.g. for section 1, in line 1of paragraph 2, a subject is missing before "has", in line 1 of paragraph 3, "except" is suggested to be replaced by "in addition to", in line 3 of paragraph 4, "It" should be smaller case and so on.

A: I'm sorry for the negligence of language. We have chosen language editing services to polish the language of the paper. The detail information of revision is shown in the first line of paragraph 3 of section 1 and so on.

(3) In the first line of section 2.3, "According to the engineering background and the similarity theory,"and "According to the similarity theory," they are in repetitive in expression.

A: Thank you for your careful review. The repetitive in expression has been revised, and the specific revision position is at the first line of the first paragraph of subsection 3.1, thank you again!

(4) The background information part needs to be shortened and the physical mechanisms lying behind the experimental phenomena should be paid more attentions to. It is more significant to dig out the reasons to explain the results.

A:Thanks for reviewer'shelping comment.

①Please allow me to make an explaintion that considering the another reviewer's suggestion to state more detail about the engineering background, so, in this paper, I add the typical field photo about the weak zone, and briefly introduce the background of the project. The add information are shown in lines 6-7 of the first paragraph of section 2 and Figure 2.

② According the constructive comment, we have made great changes to the paper, in the process of adjusting structure and logic of this paper, the corresponding test results were also sorted out. The detail information of revision of test results is shown in section 4 of the paper and section 7.

(5) The literature review need to be improved, some related newest investigations are suggested to be added.

A: Thank you for your helping comment, I agree with the comment of the reviewer very much. I have read some

the latest relevant literatures on failure modes of surrounding rock and seepage in fissures, it has great helping for my research. And they are added in paragraph 1 of introduction and the last paragraph of section 6, and list in reference list (1-7) and (52-55).

(6) The results of model tests and numerical simulations should be compared to verify each other.

A: Thank you for your helpful comments. The results of displacement field and seepage field in numerical and physical tests are compared quantitatively. The detail information of revision is shown in Figure 14 and paragraph 4 of section 4.2, and Figure 18(a) and line 6-8 of paragraph 2 of section 5.1. The results of the model test were in good agreement with those of the numerical simulation.

(7) The boundary conditions are not clear for both mechanical and seepage analysis.

A: First of all, I'm sorry that my statement of boundary of model test is not exhaustive enough. The boundary conditions for both mechanical and seepage are the same. As seen in Figure 1 above, the constraint along the X-direction was applied to both the left and right boundaries, and that along the Y-direction was applied to the front and rear boundaries. Vertical constraints along the Z-direction were applied to the bottom of the model, the surface of the model was set as a free surface.

To saturate the model with water, glass glue was used to seal the gap in the Plexiglas between the surrounding rock and the entrance to the tunnel, as shown in Figure 2 above. So, the direction of the front and back of the model is completely cut off with the atmosphere. The detail of the revised information is shown in lines 1-2 of paragraph 9 of subsection 3.3.

(8) After the numerical simulation is verified by the physical model test, Effects of the relevant parameters, such as the exposed position, the dip angle, and the number of the weak interlayers, should be further explored. This point on which the article needs to be strengthened.

A: Thank you for your helpful comments. I agree with reviewer's comment very much, as reviewer say, parameters of weak interlayer in term of the exposed position, the dip angle, and the number are important for the research of failure mechanism. This is exactly what the author wants to study in the later period. Greater attention will be accorded to this field to further examine the influence of seepage on the failure mechanism of tunnels with weak interlayers. The detail of the revised information is shown in lines 6-8 of the last paragraph of section 6. Thank you very much again!

Thank you and best regards.

Sincerely yours,

Jing Hu on behalf of the authors.

Corresponding address and mailbox: Jing Hu at College of Civil Engineering, Chongqing University, 400045, Chongqing, China, 347902268@qq.com.